# The regulation of oocyte maturation and ovulation in the closest sister group of vertebrates

Shin Matsubara[1,2], Akira Shiraishi[1], Tomohiro Osugi[1], Tsuyoshi Kawada[1], Honoo Satake[1†]*

[1]Bioorganic Research Institute, Suntory Foundation for Life Sciences, Kyoto, Japan; [2]Research Fellow of Japan Society for the Promotion of Science, Tokyo, Japan

**Abstract** Ascidians are the closest living relatives of vertebrates, and their study is important for understanding the evolutionary processes of oocyte maturation and ovulation. In this study, we first examined the ovulation of *Ciona intestinalis* Type A by monitoring follicle rupture in vitro, identifying a novel mechanism of neuropeptidergic regulation of oocyte maturation and ovulation. *Ciona* vasopressin family peptide (CiVP) directly upregulated the phosphorylation of extracellular signal–regulated kinase (CiErk1/2) via its receptor. CiVP ultimately activated a maturation-promoting factor, leading to oocyte maturation via germinal vesicle breakdown. CiErk1/2 also induced expression of matrix metalloproteinase (CiMMP2/9/13) in the oocyte, resulting in collagen degradation in the outer follicular cell layer and liberation of fertile oocytes from the ovary. This is the first demonstration of essential pathways regulating oocyte maturation and ovulation in ascidians and will facilitate investigations of the evolutionary process of peptidergic regulation of oocyte maturation and ovulation throughout the phylum Chordata.
DOI: https://doi.org/10.7554/eLife.49062.001

*For correspondence:
satake@sunbor.or.jp

Present address: †Suntory Foundation for Life Sciences, Kyoto, Japan

Competing interests: The authors declare that no competing interests exist.

## Introduction

Oocyte maturation and ovulation are critical steps in the completion of female gametogenesis in the ovary and thus for subsequent fertilization and embryogenesis (*Lane et al., 2014*). Meiosis in animal oocytes is arrested at prophase of the first division (ProI). Hormonal stimulation triggers the resumption of meiosis in most vertebrates and invertebrates, and oocytes undergo nuclear maturation upon the onset of nuclear disassembly (i.e., germinal vesicle breakdown [GVBD]). Concomitant with the progression of nuclear maturation, dynamic rearrangement of intracellular organelles promotes cytoplasmic maturation. Mature oocytes become fertilizable, and their meiosis is arrested again at a species-specific stage: metaphase of the first division (MetI, many invertebrates), metaphase of the second division (MetII, most vertebrates), or G1-phase (some echinoderms and coelenterates) until fertilization. (*Richani and Gilchrist, 2018*; *Das and Arur, 2017*; *Nagahama and Yamashita, 2008*; *Fan and Sun, 2004*; *Kishimoto, 2018*; *Von Stetina and Orr-Weaver, 2011*). Mature oocytes are then ovulated via proteolytic degradation of the follicle walls (*Richards and Ascoli, 2018*; *Takahashi et al., 2018*; *Richards and Pangas, 2010*). Some mutations in oocytes lead to infertility in the mature organism, whereas others, over a long period of time, may eventually lead to the emergence of novel species or subspecies. Therefore, the regulatory mechanisms underlying oocyte maturation and ovulation control not only the reproduction of the respective organisms but also evolutionary processes across the animal kingdom.

In vertebrates, oocyte maturation and ovulation are regulated by the hypothalamus-pituitary-gonadal axis (HPG axis); gonadotropins from the pituitary stimulate the maturation of follicles in response to hypothalamic peptide hormones, including gonadotropin-releasing hormone

(*Richani and Gilchrist, 2018*; *Das and Arur, 2017*; *Nagahama and Yamashita, 2008*; *Fan and Sun, 2004*; *Kishimoto, 2018*; *Von Stetina and Orr-Weaver, 2011*; *Richards and Ascoli, 2018*; *Takahashi et al., 2018*; *Richards and Pangas, 2010*). In contrast, neither a pituitary organ nor gonadotropins arose in most species of invertebrates, indicating the presence of HPG axis–independent regulation in these organisms. Some neuropeptides have been shown to induce oocyte maturation and ovulation (or spawning) in several species of invertebrates. The neuropeptide W/RPRPamide in jellyfish directly induces oocyte maturation and spawning (*Takeda et al., 2018*). A relaxin-like peptide stimulates the production of 1-methyladenosine, which directly induces oocyte maturation, followed by ovulation and spawning (*Mita et al., 2009*). Cubifurin (NGIWYamide) induces oocyte maturation, ovulation, and spawning in the sea cucumber (*Kato et al., 2009*). These findings demonstrate that various neuropeptides are responsible for triggering oocyte maturation and ovulation in invertebrates, and suggest that oocyte maturation and ovulation and their underlying molecular mechanisms are regulated in both a species-specific and evolutionarily conserved fashion.

Ascidians are invertebrates that belong to the Chordata superphylum as Urochordata, and thus, phylogenetically, they are the closest living relatives of vertebrates (*Delsuc et al., 2006*; *Denoeud et al., 2010*; *Satoh et al., 2014*; *Horie et al., 2018*). As such, ascidians are considered crucial model organisms in research into the evolutionary processes underlying chordate reproductive systems. Our previous studies identified a variety of vertebrate prototypic neuropeptides from the neural complex of the cosmopolitan ascidian, *Ciona intestinalis* Type A (*Ciona robusta*) (*Kawada et al., 2011*; *Brunetti et al., 2015*), and we revealed the full trajectory of visceral nerves to the ovary (*Osugi et al., 2017*). Moreover, several cognate receptors were shown to be expressed in the ovary (*Kawada et al., 2008*; *Satake et al., 2004*; *Sekiguchi et al., 2012*; *Tello et al., 2005*; *Shiraishi et al., 2019*), and some neuropeptides, such as *Ciona* tachykinin and neurotensin-like peptide 6, were found to participate in regulating pre-GVBD follicle growth (*Aoyama et al., 2008*; *Aoyama et al., 2012*; *Kawada et al., 2011*). These findings, considered in the context of the lack of a pituitary organ and gonadotropins, strongly suggest that neuropeptides also play vital roles in oocyte maturation and ovulation in *Ciona*.

*Ciona* meiosis is arrested at ProI, resumes under stimulation by an as yet unidentified factor, and is arrested again at MetI after oocyte maturation and ovulation (*Tosti et al., 2011*; *Von Stetina and Orr-Weaver, 2011*). The importance of pH and levels of cAMP and/or $Ca^{2+}$ in *Ciona* GVBD in artificial seawater (ASW) was previously demonstrated (*Silvestre et al., 2009*; *Silvestre et al., 2011*; *Tosti et al., 2011*; *Lambert, 2008*; *Lambert, 2011*). In addition, the activities of MAP kinase (MAPK) and maturation promoting factor (MPF), which are prerequisite for oocyte maturation in vertebrates, have been investigated in *Ciona* post-fertilization (*Russo et al., 1996*; *Russo et al., 1998*; *Russo et al., 2009*). However, neither the endogenous factors nor the signaling pathways regulating GVBD and ovulation in *Ciona* have been identified. Furthermore, investigations of oocyte maturation and ovulation in *Ciona* are expected to provide crucial clues for understanding the evolution and diversification of chordate reproductive systems.

In this study, we documented for the first time the process of ovulation in *Ciona* and demonstrated that the essential molecular mechanisms underlying the onset of GVBD and ovulation are regulated by *Ciona* vasopressin family peptide, CiVP.

## Results

### In vitro evaluation of *Ciona* ovulation

In contrast to oocyte maturation, ovulation in *Ciona* has yet to be documented. We therefore observed ovulation in *Ciona* in vitro. Notably, incubating *Ciona* ovaries in ASW for 16 hr resulted in the liberation of numerous follicles (*Figure 1A*). Intriguingly, time-lapse imaging revealed that follicle liberation followed the rupture of the outer follicular cell layer (*Figure 1B* and *Video 1*), and we therefore defined this phenomenon as *Ciona* ovulation. Moreover, such rupture of outer follicular cells was also observed in an isolated follicle following GVBD (*Figure 1C* and *Video 2*). These results demonstrated that both oocyte maturation and ovulation can be studied in vitro using isolated immature/pre-ovulatory follicles. These data also underscored the advantages of using *Ciona* in research examining the molecular mechanism of oocyte maturation and ovulation. Such experiments, however, will require more-effective methods for isolating immature/pre-ovulatory follicles.

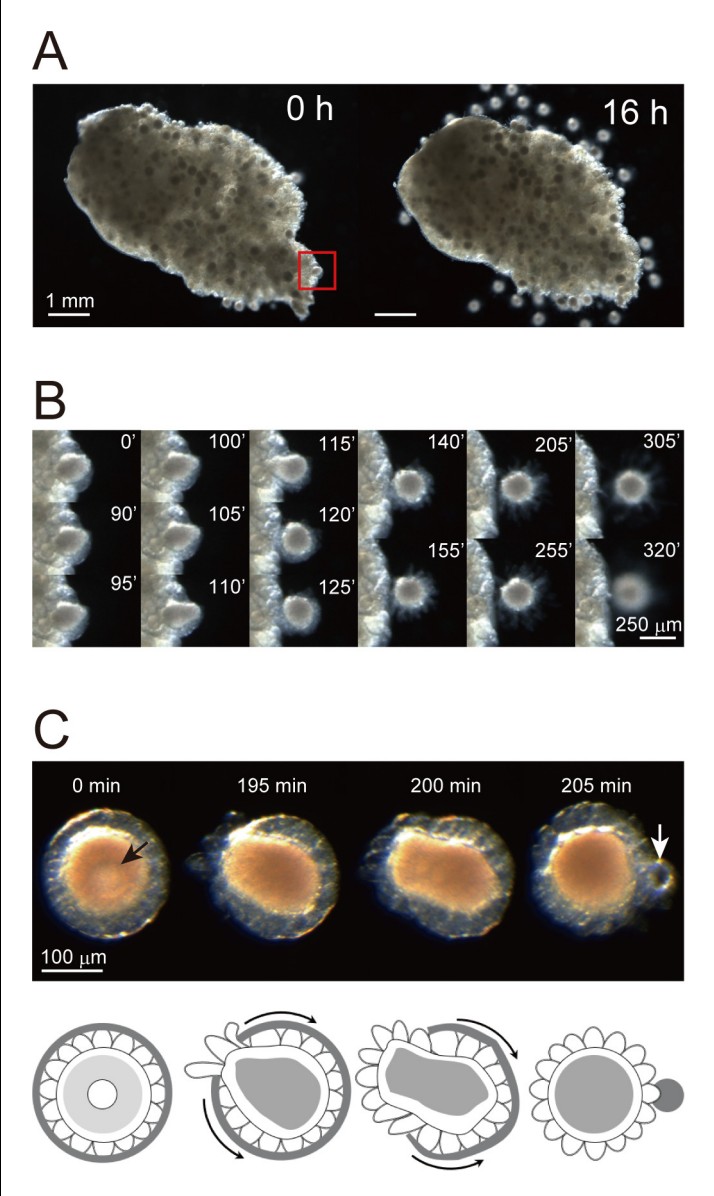

**Figure 1.** In vitro system for evaluating *Ciona* oocyte maturation and ovulation. (**A**) A *Ciona* ovary was cut into half and incubated in ASW. Numerous follicles were released from the ovary after 16 hr of incubation. (**B**) Time-lapse images of the red-boxed area in (**A**) at the indicated time. The rupturing of the outer follicular cell layer led to follicle release. (**C**) Time-lapse images of an isolated immature/pre-ovulatory (stage III) follicle at the indicated times (upper) and corresponding schematic drawings (lower). Oocyte maturation was observed with germinal vesicle (black arrow) breakdown, and the outer follicular cell layer was ruptured and shrank after ovulation (white arrow). Scale bars are shown for the indicated length.

DOI: https://doi.org/10.7554/eLife.49062.002

## Fractionation of *Ciona* follicles and whole-transcriptome analysis

In the *Ciona* ovary, follicles undergo the following major developmental stages: stage I (pre-vitellogenic), stage II (vitellogenic), stage III (post-vitellogenic and pre-GVBD), and stage IV (post-GVBD, that is, mature) oocytes (*Prodon et al., 2006*; *Tosti et al., 2011*; *Silvestre et al., 2009*). We fractionated *Ciona* follicles using stainless steel sieves of varying particle sizes. Each fraction collected (20, 38, 63, 90, or 150 μm) contained a major developmental stage of *Ciona* follicles (*Prodon et al., 2006*), namely (i) early stage I, (ii) late stage I to early stage II, (iii) early stage II to late stage II, (iv)

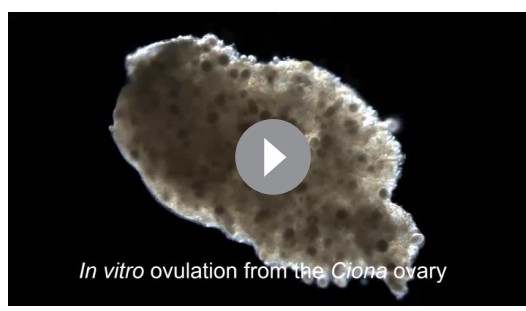

**Video 1.** In vitro ovulation of a *Ciona* ovary. An isolated *Ciona* ovary was cut in half and incubated in ASW for 16 hr. Time-lapse images were captured every 5 min. Numerous follicles were in turns ovulated from the ovary. The higher-magnification movie of the boxed area indicates that the rupturing of the outer follicular cell layer leads to follicle release.
DOI: https://doi.org/10.7554/eLife.49062.003

late stage II to late stage III, or (v) stage IV oocytes, confirming that the all developmental stages were successfully fractionated (*Figure 2A*). RNA-seq analysis of two sets of the fractionated follicles indicated no stage to stage changes in the expression of most genes; however, some differential expression of genes was detected between two consecutive stages (*Figure 2—figure supplement 1* and *Supplementary file 1*), suggesting the usefulness of this approach for identifying genes responsible for oocyte maturation and ovulation. The resultant reads, mapping rate to the *Ciona* cDNA library, and fastq accession numbers in the NCBI SRA database are listed in *Table 1*.

The immature/pre-ovulatory follicles were then divided into three groups, and the correlation between follicle stage (size) and GBVD or ovulation rate was investigated in detail after a 24 hr incubation. Small follicles (192.1 ± 9.3 µm) exhibited 12.5 ± 4.6% GVBD and 28.6 ± 4.3% ovulation. Such low rates suggested that the follicles were not yet affected by endogenous regulators (e. g., gonadotropins in vertebrates), and thus, they were categorized as 'late stage II' (*Figure 2B and C*) based on the results of a previous investigation (*Prodon et al., 2006*). In contrast, mid-size (214.3 ± 7.6 µm) and large (228.8 ± 7.0 µm) follicles exhibited 63.9 ± 8.6% and 91.8 ± 3.4% GVBD, respectively, and 61.4 ± 10.8% and 87.2 ± 1.5% ovulation, respectively, leading to their further respective categorization as 'early stage III' and 'late stage III' (*Figure 2B and C*). These late stage II, early stage III, and late stage III follicles were utilized in subsequent experiments.

## CiVP and CiErk1/2 promote both GVBD and ovulation

Expression of the appropriate receptor genes in immature/pre-ovulatory follicles is a prerequisite for the peptidergic regulation of oocyte maturation and ovulation. Among the several receptors for known *Ciona* neuropeptides in the ovary (*Kawada et al., 2008*; *Satake et al., 2004*; *Sekiguchi et al., 2012*; *Tello et al., 2005*; *Shiraishi et al., 2019*), RNA-seq analysis of fractionated follicles indicated that the expression of only the *CiVpr* gene (KH.C9.885), a receptor for CiVP, was specifically elevated, with peak expression at late stage II to late stage III, a period that is thought to be crucial in the regulation of oocyte maturation and ovulation. This expression profile for the *CiVpr* gene was confirmed by qRT-PCR analysis (*Figure 2—figure supplement 2A,B*, and *Figure 2—figure supplement 2—source data 1*). These results suggested that CiVP plays important roles in oocyte maturation and ovulation.

*Ciona* follicles are comprised of three types of cells: oocytes, test cells, and follicular cells, and in situ hybridization showed that *CiVpr* mRNA was localized to the oocytes of late stage II follicles (*Figure 3A*), in agreement with the results of RNA-seq and qRT-PCR analyses (*Figure 2—figure supplement 2B* and *Figure 2—figure supplement 2—source data 1*). These results indicated that CiVP acts on oocytes of late stage II follicles. Consequently, we evaluated the effect

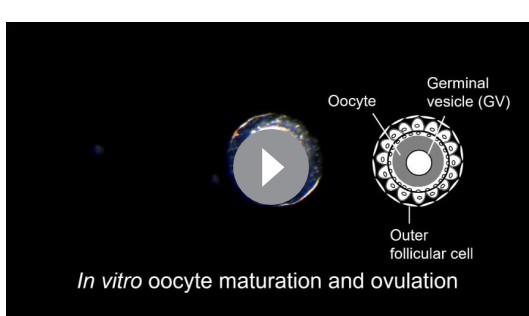

**Video 2.** In vitro oocyte maturation and ovulation of a *Ciona* follicle. An immature/pre-ovulatory (stage III) follicle was incubated in ASW for 3 hr. Time-lapse images were captured every 20 s. The follicle structure is illustrated at the right. After germinal vesicle breakdown (GVBD) occurred, the outer follicular cell layer was ruptured, and the follicle was released into ASW, reproducing the ovulation observed in the ovary (*Video 1*).
DOI: https://doi.org/10.7554/eLife.49062.004

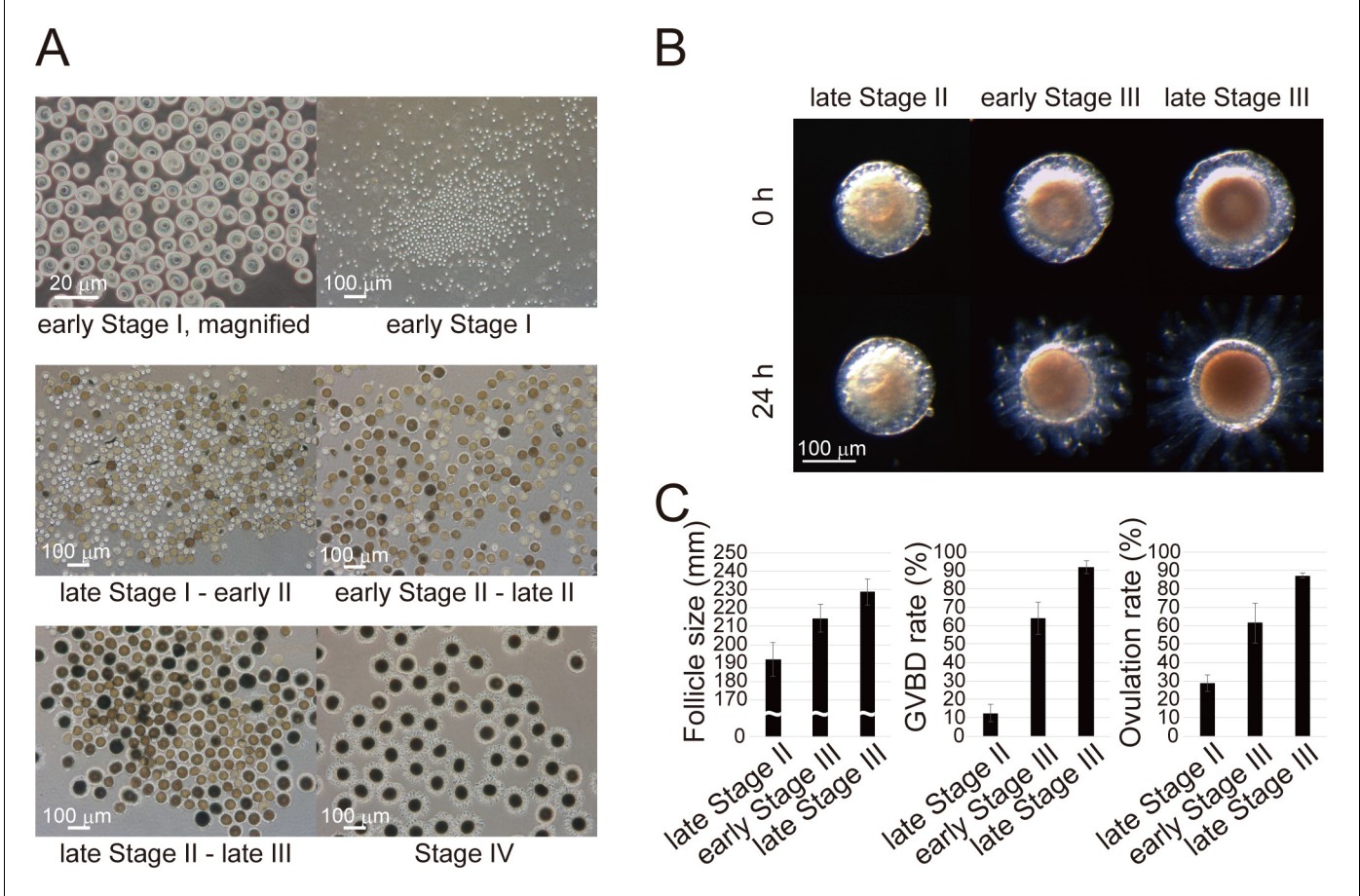

**Figure 2.** Fractionation of *Ciona* follicles. (**A**) Isolated follicles were fractionated using stainless steel sieves of varying particle sizes (20, 38, 63, 90, or 150 μm). Each fraction contained respective stage of *Ciona* follicles corresponding to (**i**) early stage I, (**ii**) late stage I to early stage II, (**iii**) early stage II to late stage II, (**iv**) late stage II to late stage III, or (**v**) stage IV oocytes. (**B**) Immature/pre-ovulatory follicles were further divided into three groups as indicated, and representative follicles are shown before and after a 24 hr incubation. Scale bars in (**A**) and (**B**) represent 20 and 100 μm. (**C**) Size, GVBD rate, and ovulation rate of late stage II (n = 145), early stage III (n = 100), and late stage III (n = 99) follicles are shown as mean ± SEM.

DOI: https://doi.org/10.7554/eLife.49062.005

The following source data and figure supplements are available for figure 2:

**Figure supplement 1.** Transcriptomic profiles during follicular development.

DOI: https://doi.org/10.7554/eLife.49062.006

**Figure supplement 2.** Expression of the *CiVpr* and *CiErk1/2* genes increases toward oocyte maturation and ovulation.

DOI: https://doi.org/10.7554/eLife.49062.007

**Figure supplement 2—source data 1.** Source data for *Figure 2—figure supplement 2*.

DOI: https://doi.org/10.7554/eLife.49062.008

of CiVP on GVBD and ovulation. Of particular significance was the finding that treating late stage II follicles with CiVP for 24 hr markedly enhanced both GVBD (39.5 ± 4.2% to 66.2 ± 4.2%) and ovulation (24.8 ± 3.7% to 72.1 ± 5.7%) compared with the control follicles (*Figure 3B,C*, *Video 3*, and *Figure 3—source data 1*). Such enhancement was not observed in the presence of inactive linear CiVP in which the cysteine residues are protected with *N*-ethylmaleimide, demonstrating the specificity of CiVP (*Figure 3B and C*). Moreover, CiVP-treated oocytes were competent for fertilization and embryonic development (*Figure 3—figure supplement 1* and *Figure 3—source data 1*), confirming that the oocytes were actually mature. Collectively, these results indicate that CiVP plays important roles in *Ciona* oocyte maturation and ovulation, which led us to further investigate the underlying molecular mechanisms.

**Table 1.** RNA-seq analysis of fractionated *Ciona* follicles.

| Sample | Total reads | % Mapped | Accession no. |
|---|---|---|---|
| **Set 1** | | | |
| early stage I | 22,071,592 | 65.72 | SRR8374812 |
| late stage I – early stage II | 21,118,716 | 90.90 | SRR8374811 |
| early stage II – late stage II | 23,935,632 | 84.31 | SRR8374810 |
| late stage II – late stage III | 23,839,575 | 50.50 | SRR8374809 |
| stage IV | 22,398,645 | 32.02 | SRR8374816 |
| **Set 2** | | | |
| early stage I | 22,317,382 | 45.99 | SRR8374815 |
| late stage I – early stage II | 24,094,643 | 88.73 | SRR8374814 |
| early stage II – late stage II | 21,434,672 | 60.23 | SRR8374813 |
| late stage II – late stage III | 20,686,040 | 43.45 | SRR8374808 |
| stage IV | 23,730,547 | 31.13 | SRR8374807 |

DOI: https://doi.org/10.7554/eLife.49062.009

In mammals, the extracellular signal-related kinase (ERK) and MAPK/ERK kinase (MEK) signaling pathways are activated by VP (*Kumari et al., 2017*; *Chiu et al., 2002*), and ERK plays a role in oocyte maturation in vertebrates (*Richani and Gilchrist, 2018*; *Das and Arur, 2017*; *Fan and Sun, 2004*; *Sha et al., 2017*). We found that the expression of *CiErk1/2* (KH.L153.20) was elevated throughout follicle development (*Figure 2—figure supplement 2C* and *Figure 2—figure supplement 2—source data 1*) and localized to oocytes and test cells during most stages, including the identical stage in *CiVpr*-expressing oocytes (*Figure 3D*). We then examined the functional relationship between CiErk1/2 and spontaneous oocyte maturation/ovulation. Inhibition of MEK/ERK activity in early stage III follicles using the MEK inhibitor U0126 resulted in significant decreases in the rates of both GVBD (86.2 ± 3.1% to 20.5 ± 9.0%) and ovulation (54.3 ± 5.8% to 6.4 ± 4.5%, *Figure 3E,F*, *Video 4*, and *Figure 3—source data 1*). Collectively, these results indicated that CiErk1/2 plays crucial roles in both oocyte maturation and ovulation and further suggest that CiVP signaling is mediated by CiErk1/2.

## CiVP promotes GVBD via the ERK-MPF pathway

Subsequently, we examined whether CiErk1/2 is phosphorylated upon CiVP stimulation. Western blotting and immunofluorescence analyses confirmed both the specificity of the anti–phosphorylated ERK1/2 (pERK1/2) antibody (*Figure 4—figure supplement 1*) and the previously reported inhibitory effect of U0126 on native CiErk1/2 phosphorylation in *Ciona* embryos (*Davidson et al., 2006*; *Chambon et al., 2002*; *Ikuta et al., 2010*; *Pasini et al., 2012*). Immunofluorescence analysis of defolliculated oocytes revealed extensive phosphorylation of CiErk1/2 at 5–10 min after CiVP stimulation, whereas no induction was observed in MEK-inhibited follicles (*Figure 4A*). RNA-seq analysis was employed to examine the expression of downstream factors of MEK/ERK signaling in oocyte maturation and ovulation using three sets of MEK-inhibited early stage III follicles (*Figure 4—figure supplement 2*, *Supplementary file 2*, and *Table 2*). We focused on the two major MPF components, cyclin B (CcnB) and Cdc2 (Cdk1), given that oocyte maturation is triggered by activation of MPF in a wide variety of both vertebrates and invertebrates (*Richani and Gilchrist, 2018*; *Nagahama and Yamashita, 2008*; *Kishimoto, 2018*; *Das and Arur, 2017*; *Fan and Sun, 2004*). BLAST analyses using the *Ciona* protein database identified the *Ciona* cyclin B (*CiCcnb*, KH.C4.213) and *Ciona* Cdc2 (also known as Cdk1, *CiCdk1*, KH.C12.372) genes. The RNA-seq data revealed no significant changes in the expression of *CiCdk1* or *CiCcnb* in MEK-inhibited follicles (*Figure 4—figure supplement 3A* and *Figure 4—source data 1*), suggesting that MPF is regulated post-translationally in *Ciona*, as in other animals (*Richani and Gilchrist, 2018*; *Nagahama and Yamashita, 2008*; *Kishimoto, 2018*; *Das and Arur, 2017*; *Fan and Sun, 2004*). The selective Cdc2 inhibitor Ro-3306 inhibited GVBD in late stage III follicles (*Figure 4B* and *Figure 4—source data 1*), confirming that MPF activity is also necessary for GVBD in *Ciona*. The inhibitory effect of Ro-3306 on native MPF

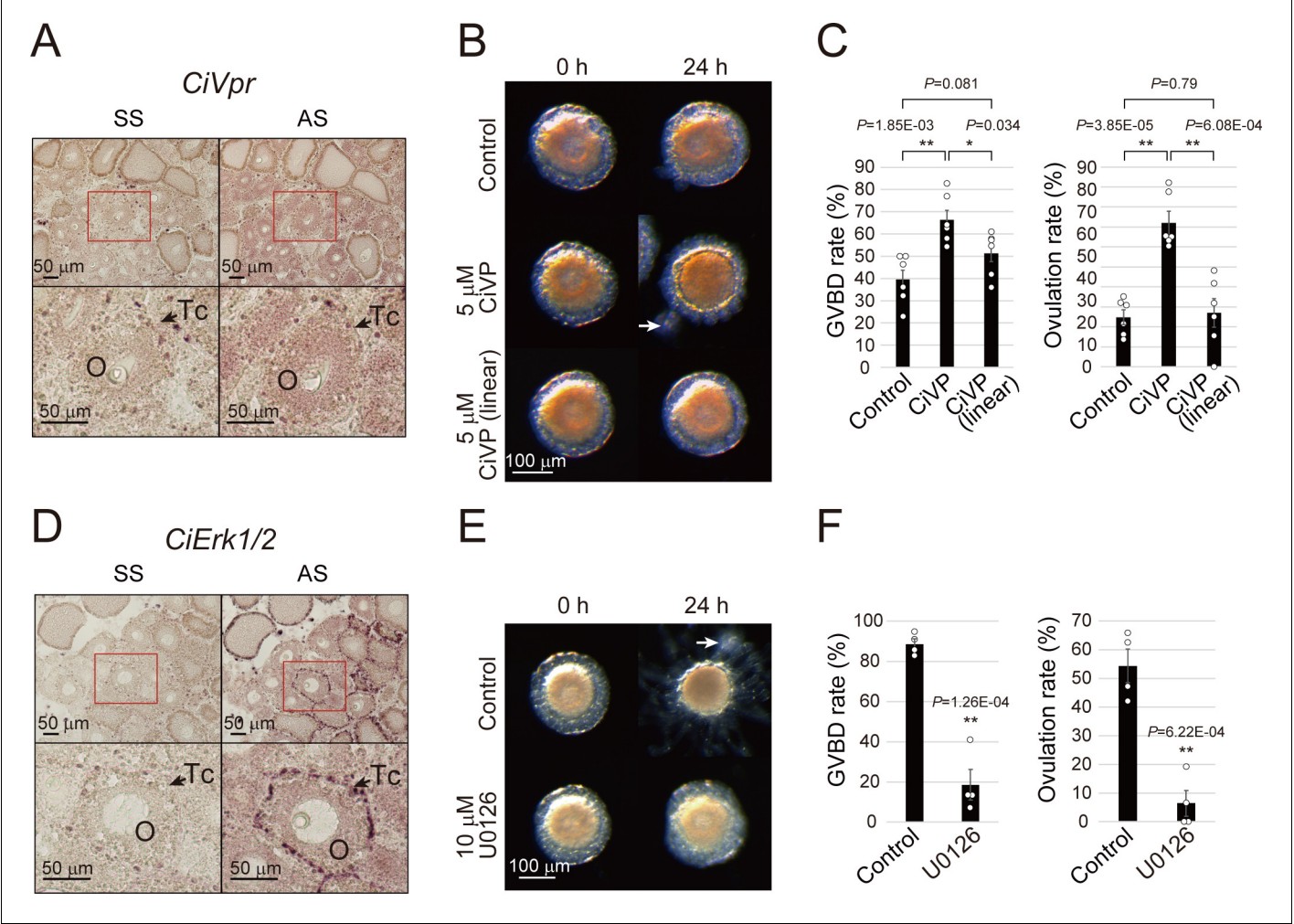

**Figure 3.** CiVP and CiErk1/2 promote both GVBD and ovulation. (**A**) In situ hybridization using sense strand (SS) or antisense strand (AS) DIG-labeled probes and an NBT/BCIP system for visualizing *CiVpr* indicated its localization in late stage II oocytes (**O**). Enlarged images of red-boxed areas in the upper panels are shown below. (**B**) Representative images of follicles before and after treatment with CiVP for 24 hr. GVBD and ovulation of late stage II follicles were enhanced in the presence of 5 µM CiVP (*t*-value: −4.19; **, p=1.84E-03 in GVBD and *t*-value: −6.97; **, p=3.85E-05 in ovulation), whereas not in the presence of 5 µM inactive linear CiVP (*t*-value: −1.94; p=0.081 in GVBD and *t*-value: −0.27; p=0.79 in ovulation). (**C**) GVBD and ovulation rates were calculated from six independent experiments (n = 6; approximately 20 follicles per experiment) and found to be significantly upregulated in CiVP-treated follicles, indicating that CiVP promotes both oocyte maturation and ovulation. Data were analyzed by Student's *t* test and are shown as mean ± SEM with data points. (**D**) In situ hybridization for *CiErk1/2* indicated its localization in oocytes (**O**) and test cells (**Tc**). (**E**) Spontaneous GVBD and ovulation of early stage III follicles were inhibited in the presence of 10 µM U0126, an MEK inhibitor. (**F**) GVBD and ovulation rates from four independent experiments (n = 4) indicated that CiErk1/2 regulates both oocyte maturation (*t*-value: 8.72; **, p=1.26E-04) and ovulation (*t*-value: 6.52; **, p=6.22E-04). Data were analyzed and are shown as in (**C**). White arrows (**B, E**) indicate outer follicular cells remaining alongside the follicle. Scale bars in (**A, D**) and (**B, E**) represent 50 and 100 µm, respectively.

DOI: https://doi.org/10.7554/eLife.49062.010

The following source data and figure supplement are available for figure 3:

**Source data 1.** Source data for *Figure 3C,F*, and *Figure 3—figure supplement 1*.
DOI: https://doi.org/10.7554/eLife.49062.012

**Figure supplement 1.** CiVP-treated follicles are competent for fertilization and development.
DOI: https://doi.org/10.7554/eLife.49062.011

(CiCdk1) was confirmed using an in vitro assay (*Figure 4—figure supplement 3B,C*, and *Figure 4— source data 1*). Notably, CiVP-directed GVBD using late stage II follicles was completely blocked by MPF inhibition (*Figure 4C* and *Figure 4—source data 1*), indicating that MPF is required for the

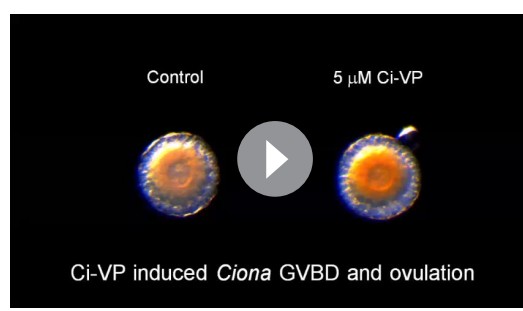

**Video 3.** CiVP induces *Ciona* GVBD and ovulation. Isolated late stage II follicles were incubated with (right) or without (left) 5 μM CiVP for 24 hr. Time-lapse images were captured every 5 min. GVBD and ovulation were observed in the CiVP-treated follicle, whereas not in the control follicle.
DOI: https://doi.org/10.7554/eLife.49062.013

induction of GVBD by CiVP. Consequently, these results led to the conclusion that CiVP is a key molecule in the initiation of oocyte maturation via the MEK/ERK-MPF pathway in *Ciona* oocytes.

## CiVP activates CiMMP2/9/13 via MEK/ERK signaling prior to ovulation

In medaka, a teleost fish species, three MMPs (MMP-2, MMP-14, MMP-15) have been identified as essential enzymes for ovulation (*Ogiwara et al., 2005*), but no proteases have been identified as essential for mammalian ovulation. In *Ciona*, six MMP-like genes were annotated with KH accession information in the *Ciona* gene database (*Kawai et al., 2015*), and thus, the expression profiles of these genes were examined using the present RNA-seq data (*Figure 5—figure supplement 1*) for early stage III follicles in which ovulation was inhibited with an MEK inhibitor. Of these six genes, the expression of only *CiMmp2/9/13* (KH.L76.4) declined by approximately 80% in MEK-inhibited follicles compared with the control (*Figure 5—figure supplement 1* and *Figure 5—source data 1*). The results of qRT-PCR analysis using MEK-inhibited follicles also revealed a significant 70% decrease in *CiMmp2/9/13* mRNA levels (*Figure 5A* and *Figure 5—source data 1*). Notably, CiVP stimulation of late stage II follicles resulted in a 4.0-fold upregulation of *CiMmp2/9/13* expression (*Figure 5B* and *Figure 5—source data 1*), confirming that CiVP induces *CiMmp2/9/13* expression via MEK/ERK signaling during ovulation. This led us to further investigate the functional roles of CiMMP2/9/13 protein in *Ciona* ovulation.

We generated an anti-CiMMP2/9/13 antibody using the active form of recombinant CiMMP2/9/13 (rCiMMP2/9/13), and the antibody's specificity was confirmed by Western blotting analysis (*Figure 5—figure supplement 2*). Immunohistochemistry analysis revealed that CiMMP2/9/13 protein localized in oocytes and the outer follicular cell layer of late stage III follicles (*Figure 5C*), and this result was in good agreement with the localization of CiVPR (*Figure 3A*) and CiErk1/2 (*Figure 3D*). Subsequently, we treated late stage III follicles with MMP-2/9 inhibitor II. As expected, inhibition of CiMMP2/9/13 resulted in a significant decrease in the ovulation rate (91.5 ± 3.2% to 41.4 ± 3.1%), compared with uninhibited follicles (*Figure 5D* and *Figure 5—source data 1*). It is noteworthy that CiVP-induced ovulation via upregulation of CiMMP2/9/13 in late stage II follicles was completely blocked by MMP-2/9 inhibitor II (*Figure 5E* and *Figure 5—source data 1*). Collectively, these results indicated that CiMMP2/9/13 expression is induced by CiVP via CiErk1/2 in oocytes and plays a pivotal role in *Ciona* ovulation.

## CiMMP2/9/13 digests both collagen type I and type IV around the outer follicular cell layer

MMPs digest extracellular matrix components, including collagens (*Löffek et al., 2011*), and our current data support the hypothesis that CiMMP2/9/13 digests collagen type I and type IV in *Ciona* follicles during ovulation. Incubation of rCiMMP2/9/13 with putative substrates DQ collagen (fluorescent dye and its quencher are

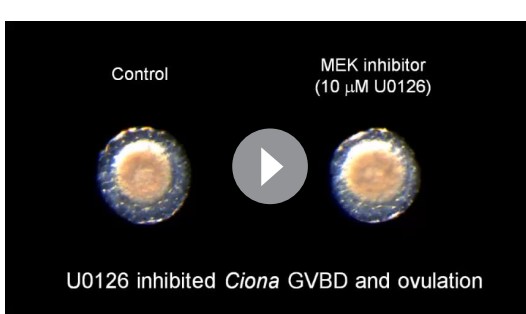

**Video 4.** The MEK inhibitor U0126 inhibits *Ciona* GVBD and ovulation. Isolated early stage III follicles were incubated with (right) or without (left) 10 μM U0126 for 24 hr. Time-lapse images were captured every 5 min. GVBD and ovulation were observed in the control follicle, whereas those were inhibited in the U0126-treated follicle.
DOI: https://doi.org/10.7554/eLife.49062.014

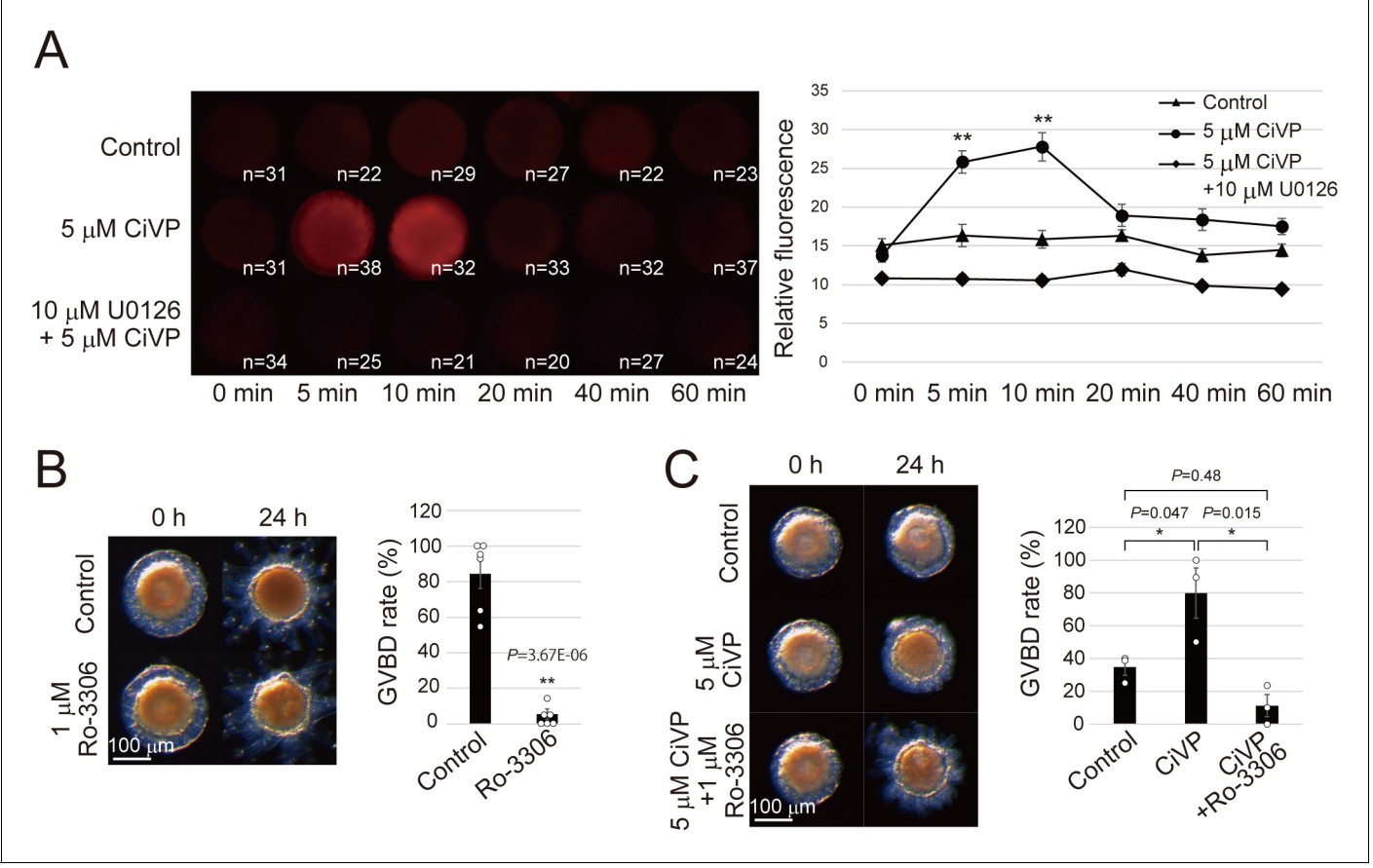

**Figure 4.** CiVP promotes GVBD via CiErk1/2-CiMPF pathway. (**A**) Indicated numbers of numbers of de-folliculated oocytes pretreated with or without 10 µM U0126 were stimulated with 5 µM CiVP for 0, 5, 10, 20, 40, and 60 min. Immunofluorescence using anti-phosphorylated ERK1/2 antibody indicated specific activation of CiErk1/2 at 5 to 10 min after CiVP stimulation ($F = 14.91$; p=2.12E-12), which was not observed in control ($F = 1.05$; p=0.39) or U0126-pretreated oocytes ($F = 2.13$; p=0.065). Representative images are shown (left). Signal intensities were quantified using Fiji software (right), and significant activation was verified by one-way factorial ANOVA followed by Tukey's *post hoc* test (**, $P<0.01$). (**B**) Treatment with 1 µM Ro-3306, a selective Cdc2 inhibitor, significantly inhibited GVBD in late stage III follicles (t-value: 9.12; **, p=3.67E-06). (**C**) CiVP-induced GVBD of late stage II follicles (t-value: −2.83; *, p=0.047) was disrupted by Ro-3306 (t-value: 4.12; *, p=0.015). Representative images of follicles and GVBD rates are shown. Data from six (B, n = 6) and three (C, n = 3) independent experiments were analyzed by Student's *t* test and are shown as mean ± SEM with data points.

DOI: https://doi.org/10.7554/eLife.49062.015

The following source data and figure supplements are available for figure 4:

**Source data 1.** Source data for *Figure 4B,C*, *Figure 4—figure supplement 3A and C*.
DOI: https://doi.org/10.7554/eLife.49062.019
**Figure supplement 1.** Specificity of the anti-pERK1/2 antibody.
DOI: https://doi.org/10.7554/eLife.49062.016
**Figure supplement 2.** DEG-profile in MEK-inhibited follicles.
DOI: https://doi.org/10.7554/eLife.49062.017
**Figure supplement 3.** *Ciona* MPF is responsible for CiVP-directed oocyte maturation.
DOI: https://doi.org/10.7554/eLife.49062.018

conjugated) type I or type IV in vitro resulted in significant digestion of these collagens, but collagen digestion was inhibited by MMP-2/9 inhibitor II. These results confirmed the enzymatic activity of CiMMP2/9/13 and the usefulness of its inhibitor (*Figure 5—figure supplement 3* and *Figure 5—source data 1*).

Next, we performed in situ zymography analysis to determine the localization of CiMMP2/9/13 activity within *Ciona* follicles during and after ovulation. While no activity was observed in oocytes, test cells, and follicular cells in ovulating follicles, potent collagenase activity was observed in the

**Table 2.** RNA-seq analysis of MEK-inhibited follicles

| Sample | Total reads | % Mapped | Accession no. |
|---|---|---|---|
| Set 1 | | | |
| Control | 32,160,880 | 74.49 | SRR8375763 |
| U0126 | 24,928,350 | 74.46 | SRR8375762 |
| Set 2 | | | |
| Control | 32,642,355 | 72.66 | SRR8375761 |
| U0126 | 32,197,268 | 64.87 | SRR8375760 |
| Set 3 | | | |
| Control | 32,136,047 | 58.70 | SRR8375759 |
| U0126 | 25,912,488 | 60.77 | SRR8375758 |

DOI: https://doi.org/10.7554/eLife.49062.020

outer follicular cell layer in the presence of DQ collagen type I or type IV (*Figure 6*). Intriguingly, the outer follicular layer of post-ovulatory follicles also exhibited high activity toward collagen type I and type IV. Collectively, these results led to the conclusion that CiVP induces CiMMP2/9/13 expression via CiVPR-MEK/ERK signaling in oocytes and that secreted CiMMP2/9/13 digests collagens type I and type IV in the outer follicular cell layer, which causes the follicle to rupture, with subsequent liberation of oocytes from the ovary (i.e., ovulation).

## Discussion

Oocyte maturation and ovulation are regulated by the HPG axis in vertebrates, whereas most invertebrates do not have an HPG axis. Hence, invertebrate reproductive processes are regulated by HPG axis–independent systems, but many details of the underlying molecular mechanisms of these systems remain to be elucidated. Over the past decade, some neuropeptides were found to play key roles in oocyte maturation and ovulation (or spawning) in a limited number of invertebrates, such as jellyfish (W/RPRPamide, *Takeda et al., 2018*), starfish (relaxin-like peptide, *Mita et al., 2009*), and sea cucumber (cubifrin: NGIWYamide, *Kato et al., 2009*). In contrast, no endogenous factors involved in the regulation of oocyte maturation or ovulation have been identified in ascidians. This has hampered investigations into the molecular mechanisms underlying the reproductive system in the closest sister group to vertebrates and the evolution and diversification of the oocyte maturation and ovulation systems in chordates. Using original assays for examining ovulation in *Ciona* follicles (*Figures 1* and *2*) in the present study, we demonstrated that essential systems in oocyte maturation and ovulation in the ascidian *Ciona intestinalis* Type A are regulated by the vasopressin/oxytocin (VP/OT) family peptide, CiVP (*Figure 7*).

One of the greatest difficulties in studies of oocyte maturation and ovulation involves the isolation and fractionation of vital follicles and oocytes at each developmental stage. In this study, we established a method for culturing and fractionating vital *Ciona* follicles across all developmental stages using stainless steel sieves of varying particle sizes (*Figure 2A*). This method enabled both the first observation of *Ciona* ovulation (*Figure 1C* and *Video 2*) and exploration of the molecular mechanisms underlying oocyte maturation and ovulation (*Figures 3–7*). Such fractionation and culture systems are likely to be applicable to other organisms and thus should encourage studies of oocyte maturation and ovulation in various species. Moreover, transcriptomic data covering all developmental stages of *Ciona* follicles (*Figure 2A*, *Figure 2A—figure supplement 1*, and *Supplementary file 1*) will contribute to the comprehensive understanding of not only oocyte maturation and ovulation but also the entire mechanism of *Ciona* follicle development.

The vertebrate OT and VP families are thought to have emerged from a common ancestral gene via gene duplication during the evolution of gnathostomates, and invertebrates and jawless vertebrates have been shown to possess a single VP/OT family peptide (*Banerjee et al., 2017*; *Stoop, 2012*; *Donaldson and Young, 2008*). Furthermore, the involvement of VP/OT family peptides in various reproduction-related functions has been documented. The earthworm VP/OT family peptide annetocin and nematode VP/OT family peptide nematocin induce egg-laying behavior

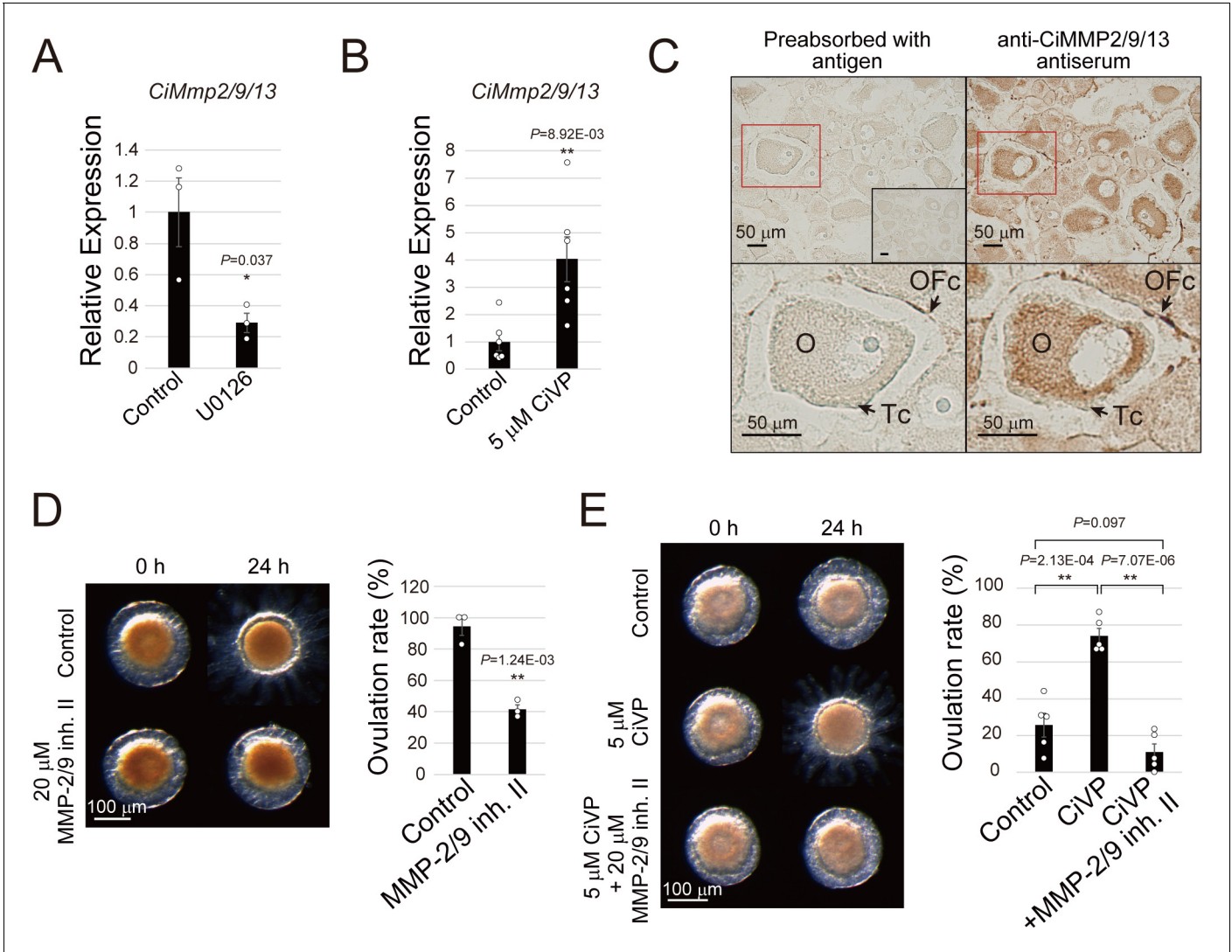

**Figure 5.** CiVP activates CiMMP2/9/13 via MEK/ERK signaling during ovulation. (**A, B**) qRT-PCR indicated that MEK inhibition with 10 µM U0126 in early stage III follicles (**A**) and 5 µM CiVP stimulation in late stage II follicles (**B**) for 24 hr results in decrease and increase of *CiMmp2/9/13* expression relative to *CiUbac1*, respectively. Data from three (A, n = 3) and six (B, n = 6) independent experiments were analyzed by Student's *t* test and are shown as mean ± SEM with data points (*t*-value: 3.08; *, p=0.037 in (**A**) and *t*-value: −3.24; **, p=8.92E-03 in (**B**)). (**C**) Immunohistochemistry using anti-CiMMP2/9/13 antiserum and the ABC detection system confirmed that CiMMP2/9/13 is localized in oocytes (**O**) and outer follicular cells (**OFc**), but not in test cells (Tc). Antiserum pre-absorbed with antigen or pre-immune serum (inset) were used as negative controls. Enlarged images of red-boxed areas in the upper panels are shown below. (**D**) Treatment with 20 µM MMP-2/9 inhibitor II significantly inhibited ovulation of late stage III follicles (*t*-value: 8.14; **, p=1.24E-03). (**E**) Significant CiVP-induced ovulation of late stage II follicles (*t*-value: −6.38; **, p=2.13E-04) was disrupted by MMP-2/9 inhibitor II (*t*-value: 10.25; **, p=7.07E-06). Data from three (D, n = 3) and five (E, n = 5) independent experiments were analyzed by Student's *t* test and are shown as mean ± SEM with data points.

DOI: https://doi.org/10.7554/eLife.49062.021

The following source data and figure supplements are available for figure 5:

**Source data 1.** Source data for *Figure 5A,B,D,E*, *Figure 5—figure supplements 1* and *3*.
DOI: https://doi.org/10.7554/eLife.49062.025
**Figure supplement 1.** *Ciona*-MMP expressions in MEK-inhibited follicles.
DOI: https://doi.org/10.7554/eLife.49062.022
**Figure supplement 2.** Specificity of the anti-CiMMP2/9/13 antibody.
DOI: https://doi.org/10.7554/eLife.49062.023
**Figure supplement 3.** In vitro collagenase assay.
DOI: https://doi.org/10.7554/eLife.49062.024

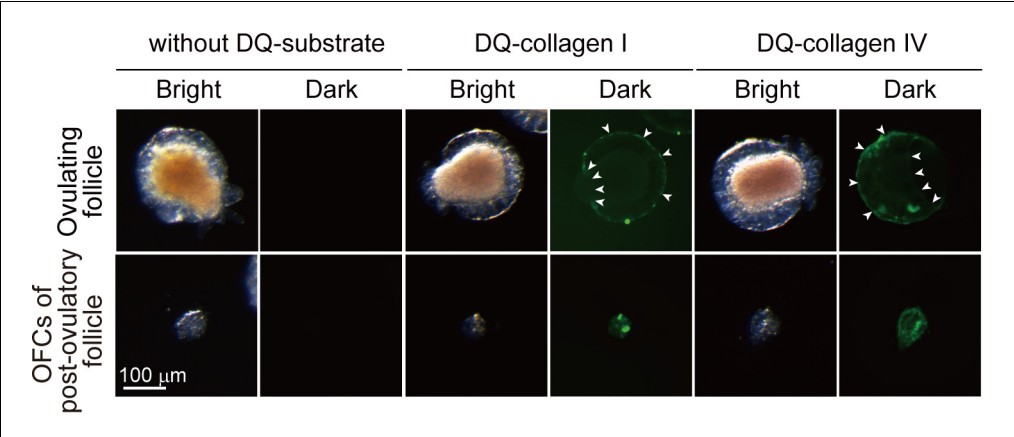

**Figure 6.** Collagenase activity is localized to the outer follicular cells. Ovulating follicles and outer follicular cells of post-ovulatory follicles were incubated for 3 hr in the presence of 50 µg/ml of DQ-collagen type I or type IV. In situ zymography indicated that the collagenase activity was localized to the outer follicular cells of ovulating follicles (upper, arrowheads) and post-ovulatory follicles (lower). No signals were observed in the follicles incubated without DQ-substrates.
DOI: https://doi.org/10.7554/eLife.49062.026

(*Fujino et al., 1999*; *Oumi et al., 1996*; *Satake et al., 1999*; *Garrison et al., 2012*). Vasotocin (VT), a non-mammalian vertebrate VP family peptide exhibited similar functions in amphibians (*Moore et al., 1992*), and its involvement in spawning migration in fish and ovipositioning in birds was also suggested, although the underlying molecular mechanisms remain to be investigated. VT was also reported to be involved in oocyte maturation and ovulation in catfish (*Singh and Joy, 2011*; *Joy and Chaube, 2015*). However, VT-induced oocyte maturation and ovulation are mediated primarily via the VT receptor in follicular cells and following sexual steroidogenesis (*Singh and Joy, 2011*; *Joy and Chaube, 2015*; *Rawat et al., 2019*). This is distinct from the present data indicating that CiVP directly acts on CiVPR expressed in oocytes to induce oocyte maturation and ovulation without steroidogenesis. This is also consistent with the complete absence of major steroidogenesis genes in the *Ciona* genome (*Dehal et al., 2002*). Collectively, these findings suggest that VT and CiVP participate in oocyte maturation and ovulation via distinct molecular mechanisms in catfish and *Ciona*, respectively. In other words, the biological role of VP family peptides in oocyte maturation and ovulation might have been acquired at least by common ancestors of Olfactores (vertebrates and urochordates) and conserved in both *Ciona* and fish, with diversification of the molecular mechanisms in species-specific evolutionary lineages. In mammals, the murine VP-receptor genes *Avpr1a* and *1b* and OT-receptor gene *Oxtr* are reportedly expressed in the ovary (MGI Direct Data Submission MGI:3625434, MGI:3625077, MGI:5895306, and SRS307248). However, no effects of deficiencies of these genes on oocyte maturation and ovulation in mice have been reported (*Koshimizu et al., 2012*; *Takayanagi et al., 2005*). Taken together, these results suggest that the biological roles of VPergic oocyte maturation and ovulation might have been lost or replaced by other factors in mammals. Consequently, elucidating the biological roles of VP/OT family peptides in the ovary and in the process of VPergic oocyte maturation and ovulation evolution awaits further investigations.

Oocyte maturation (GVBD) in vertebrates (meiotic resumption from ProI to MetII via GVBD) is regulated in follicles by sex steroids and subsequently by several signaling cascades, including the Mos-MAPK and MPF pathways (*Richani and Gilchrist, 2018*; *Das and Arur, 2017*; *Nagahama and Yamashita, 2008*; *Fan and Sun, 2004*). In ascidians, MAPK and MPF activity function after fertilization, given that *Ciona* oocytes are re-arrested at MetI after GVBD and resume meiosis in response to fertilization (*Russo et al., 1996*; *Russo et al., 1998*; *Russo et al., 2009*; *Dumollard et al., 2011*). However, the functional roles of MAPK and MPF at the onset of *Ciona* GVBD and the identity of their upstream regulator remain unclear. We demonstrated that CiErk1/2 is directly phosphorylated by CiVP in the oocytes (*Figure 4A*) and that MEK inhibition prevents spontaneous GVBD of early

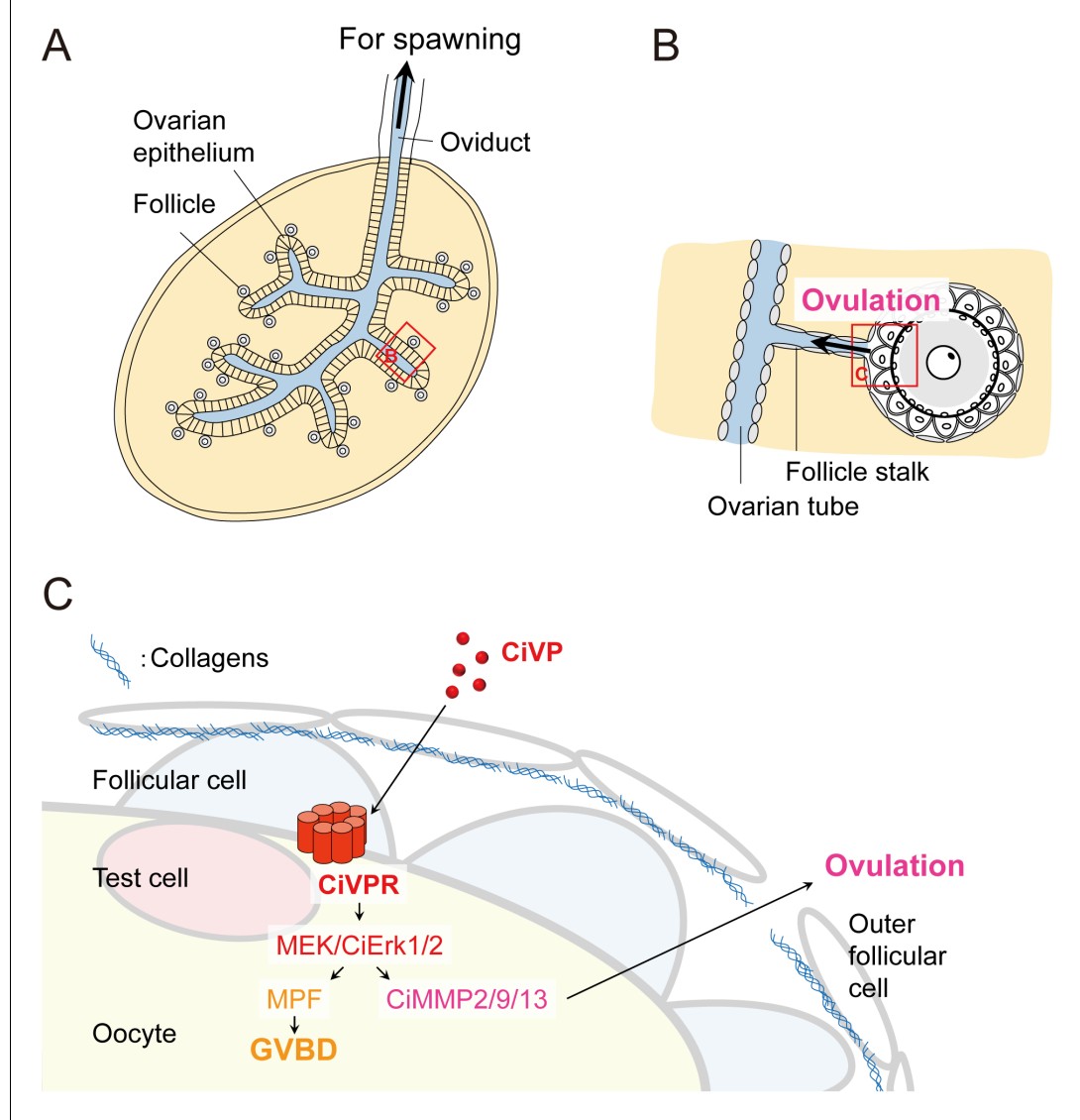

**Figure 7.** Model describing the regulation of CiVP-directed oocyte maturation and ovulation. (**A**) Schematic diagram of the *Ciona* ovary, modified from *Sugino et al. (1990)*. Mature/ovulated follicles are stored in the oviduct and wait for spawning stimulation. Yellow and blue represent inside and outside of the ovary, respectively. (**B**) Enlarged structure of ovulating follicle indicated by the red box in (**A**), modified from *Kawamura et al. (2011)*. (**C**) Enlarged illustration of the red box in (**B**) and regulatory mechanisms of the CiVP-directed *Ciona* oocyte maturation and ovulation. CiVP interacts with its receptor expressed in the oocyte and directly activates MEK/CiErk1/2 signaling. Subsequent activation of CiMPF leads to GVBD and oocyte maturation. Activated CiErk1/2 also induces the CiMMP2/9/13 expression in the oocyte; secreted CiMMP2/9/13 then digests collagens in the outer follicular cell layer, leading to the rupture of the follicular layer and ovulation.

DOI: https://doi.org/10.7554/eLife.49062.027

The following figure supplement is available for figure 7:

**Figure supplement 1.** Schematic model of the evolution of ovulatory systems.
DOI: https://doi.org/10.7554/eLife.49062.028

stage III follicles (*Figure 3E and F*), in contrast to a previous study (*Lambert, 2008*). This discrepancy is likely the result of the involvement of CiErk1/2 in GVBD of *Ciona* oocytes over a limited period; CiErk1/2 is triggered around early stage III; subsequent MEK inhibition during late stage III is no longer able to prevent spontaneous GVBD. Although CiErk1/2 is expressed both in oocytes and test cells (*Figure 3D*), de-folliculated *Ciona* oocytes are competent for GVBD (*Silvestre et al., 2009*), supporting the hypothesis that CiErk1/2 in oocytes (rather than in test cells) is important for *Ciona* GVBD, as reported in *Xenopus* (*Kosako et al., 1994*; *Cross and Smythe, 1998*; *Fan and Sun,*

*2004*). Moreover, we verified that spontaneous GVBD of late stage III follicle is disrupted by inhibition of a major component of MPF, Cdc2, indicating that MPF activity is responsible for *Ciona* GVBD, as reported in vertebrates (*Das and Arur, 2017*; *Nagahama and Yamashita, 2008*), starfish (*Kishimoto, 2018*), fruit flies (*Von Stetina et al., 2008*; *Hong et al., 2003*), and nematodes (*Burrows et al., 2006*). In addition, we demonstrated that CiVP-induced GVBD is completely blocked by MPF (Cdc2) inhibition (*Figure 4C*), which is consistent with the observation that the triggering factors of oocyte maturation (e.g., maturation-inducing hormone) are highly variable among species (*Takeda et al., 2018*; *Kishimoto, 2018*; *Nagahama and Yamashita, 2008*). Collectively, the current results provide evidence that CiVP triggers *Ciona* nuclear maturation of oocytes via the MEK/ERK-MPF pathway, which appears to be highly conserved among some vertebrates. Additionally, in vitro fertilization experiments using follicles in which maturation was induced by CiVP showed incomplete development of some follicles (*Figure 3—figure supplement 1*), suggesting the involvement of other factors in nuclear or cytoplasmic maturation of oocytes. However, the present study provides new evidence identifying CiVP as the only VP/OT family peptide that plays a crucial role in initiating GVBD via the MEK/ERK-MPF pathway in a chordate. Thus, how VP was replaced by other factors in MEK/ERK-MPF–directed GVBD in vertebrates is of particular interest.

Neither ovulation itself nor its mechanism have been documented in *Ciona*. We verified that the VPergic ovulation process involves CiVP-mediated upregulation of CiMMP2/9/13 via CiErk1/2 activation, in turn leading to the degradation of collagens followed by the release of oocytes from the ovary (*Figures 3*, *4A*, *5* and *6*). This is in good agreement with the results of previous studies of medaka demonstrating that MMP-2 and MMP-15 are responsible for the respective degradation of type IV and type I collagens and ovulation (*Ogiwara et al., 2005*; *Takahashi et al., 2018*), although medaka MMP-15 is induced by sex steroids (*Ogiwara and Takahashi, 2017*) and no role for medaka ERK in ovulation has been demonstrated. In contrast, conditional ERK1/2-deficient mice lack ovulated oocytes (*Fan et al., 2009*), and critical proteases have yet to be identified. Consequently, the present study provides the first demonstration of the biological roles of VP family peptides in regulating ovulation and inducing MMP via ERK in chordates (*Figure 7—figure supplement 1*).

Considered along with the aforementioned involvement of MMPs in ovulation in vertebrates, the present data regarding the MEK/ERK-MMP regulatory pathway in *Ciona* ovulation suggest the following three-step evolutionary process: i) common chordate ancestors might have adopted the MEK/ERK-MMP pathway, which is also conserved in *Ciona* ovulation; ii) MEK/ERK regulation in fish might have been lost or compensated by other regulators, given that an MEK/ERK-directed ovulatory mechanism has been demonstrated in mammals; and iii) the roles of various proteases, including MMPs, in vertebrate ovulatory regulation have diversified during the evolutionary process (*Figure 7—figure supplement 1i-iii*). Although the evolutionary conservation and detailed mechanism of ovulatory regulation via the MEK/ERK-MMP pathway in chordates await further investigation, the results of the present study suggest that VP/OT family peptide–directed regulation of oocyte maturation and ovulation might have originated in ancestral chordates and that other regulators, including sex steroids, prostaglandins, and gonadotropins, diverged along with the evolution of vertebrates.

In conclusion, the present study provides evidence that CiVP triggers oocyte maturation and ovulation via MEK/ERK signaling followed by MPF- and CiMMP2/9/13-directed collagen degradation, respectively, in the closest sister group of vertebrates. These data suggest that VP/OT family peptides have played key roles in the evolution of regulatory mechanisms underlying oocyte maturation and ovulation in chordates.

## Materials and methods

**Key resources table**

| Reagent type (species) or resource | Designation | Source or reference | Identifiers | Additional information |
| --- | --- | --- | --- | --- |
| Gene (*Ciona intestinalis* type A) | *CiVp* | *Ciona* Ghost Database (http://ghost.zool.kyoto-u.ac.jp/cgi-bin/gb2/gbrowse/kh/) | KH.C6.11. | |

*Continued on next page*

*Continued*

| Reagent type (species) or resource | Designation | Source or reference | Identifiers | Additional information |
|---|---|---|---|---|
| Gene (*Ciona intestinalis* type A) | *CiVpr* | *Ciona* Ghost Database (http://ghost.zool.kyoto-u.ac.jp/cgi-bin/gb2/gbrowse/kh/) | KH.C9.885. | |
| Gene (*Ciona intestinalis* type A) | *CiErk1/2* | *Ciona* Ghost Database (http://ghost.zool.kyoto-u.ac.jp/cgi-bin/gb2/gbrowse/kh/) | KH.L153.20. | |
| Gene (*Ciona intestinalis* type A) | *CiCcnb* | *Ciona* Ghost Database (http://ghost.zool.kyoto-u.ac.jp/cgi-bin/gb2/gbrowse/kh/) | KH.C4.213. | |
| Gene (*Ciona intestinalis* type A) | *CiCdk1* | *Ciona* Ghost Database (http://ghost.zool.kyoto-u.ac.jp/cgi-bin/gb2/gbrowse/kh/) | KH.C12.372. | |
| Gene (*Ciona intestinalis* type A) | *CiMmp2/9/13* | *Ciona* Ghost Database (http://ghost.zool.kyoto-u.ac.jp/cgi-bin/gb2/gbrowse/kh/) | KH.L76.4. | |
| Recombinant DNA reagent | pET21a-CiMMP2/9/13 (plasmid) | This paper | | Expression vector of Ci-MMP2/9/13 |
| Strain, strain background (*Escherichia coli*) | Rossetta 2 (DE3) | Novagen | Cat.#:71400–3 | Competent cells for rCi-MMP2/9/13 expression |
| Antibody | anti-ERK (rabbit polyclonal) | Cell Signaling Technology | Cat.#:9102S; RRID:AB_330744 | WB (1:500) |
| Antibody | anti-pERK (rabbit polyclonal) | Cell Signaling Technology | Cat.#:9101S; RRID:AB_331646 | IF (1:200); WB (1:1000) |
| Antibody | anti-PSTAIR (mouse monoclonal) | Abcam | Cat.#:Ab10345; RRID:AB_297080 | WB (1:1000) |
| Antibody | anti-CiMMP2/9/13 | This paper | | IHC (1:400); WB (1:3000) |
| Commercial assay or kit | MESACUP Cdc2/Cdk1 Kinase Assay kit | MBL | Cat.#:5235 | |
| Commercial assay or kit | Universal Elite ABC kit | Vector Laboratories | Cat.#:PK-7200 | |
| Peptide, recombinant protein | CiVP | *Kawada et al., 2008* (PMID: 18586058) | | 5 µM |
| Chemical compound, drug | U0126 | Promega | Cat.#:V1121 | 10 µM |
| Chemical compound, drug | Ro-3306 | Abcam | Cat.#:ab141491 | 1 µM |
| Chemical compound, drug | MMP-2/MMP-9 inhibitor II | Calbiochem | Cat.#:444249 | 20 µM |
| Chemical compound, drug | DQ Collagen, type I | Thermo Fisher Scientific | Cat.#:D12060 | 50 µg/ml |
| Chemical compound, drug | DQ Collagen, type IV | Thermo Fisher Scientific | Cat.#:D12052 | 50 µg/ml |
| Software, algorithm | Free-JSTAT statistical software | FreeJSTAT (http://toukeijstat.web.fc2.com/) | | Version 22.0E |
| Software, algorithm | Fiji software | Fiji (http://fiji.sc/); *Schindelin et al., 2012* (PMID: 22743772) | RRID:SCR_002285 | |
| Sequence-based reagent | Primers | This paper | | Sequences are listed in *Table 3* |

## In vitro oocyte maturation and ovulation of *Ciona* follicles

Adult *Ciona* were cultivated at the Maizuru Fisheries Research Station of Kyoto University or Misaki Marine Biological Station of the University of Tokyo and maintained in sterile ASW at 18°C. Half pieces of the ovaries or follicles from adult *Ciona* were incubated at 20°C; time-lapse images were captured every 20 s or 5 min over a 3-, 16-, or 24 hr period using a Leica M205 AF stereomicroscope. The *Ciona* ovary includes numerous immature/pre-ovulatory follicles beside the ovarian tube, which leads to the oviduct. Mature oocytes in the ovary are extruded into the ovarian tube (ovulated) and stored in the oviduct until spawning (*Figure 7A and B*; *Sugino et al., 1990*; *Kawamura et al., 2011*). Oocyte maturation was evaluated as an index of GVBD. Oocyte maturity was examined using an in vitro fertilization assay (*Figure 3—figure supplement 1*), the detailed method of which is described below. Ovulation was defined as rupturing of the outer follicular cell layer followed by release of the ovulated follicles. Rates of GVBD and ovulation were calculated 24 hr after incubation in the presence or absence of various chemicals and/or neuropeptides. Follicular size was measured using Fiji software (*Schindelin et al., 2012*).

Isolated *Ciona* follicles were fractionated using a series of stainless steel sieves of varying particle sizes (150, 90, 63, 38, and 20 µm). Thousands of follicles were collected from each sieve after washing with an excess amount of ASW. Two independent sets of follicles were used for RNA-seq, and three additional sets were used for qRT-PCR. Immature/pre-ovulatory follicles were further isolated under a stereomicroscope (Discovery V8, Carl Zeiss Microscopy, Tokyo, Japan) and divided into three gropes (late stage II, early stage III, and late stage III) based on size, and GVBD and ovulation rates were determined after a 24 hr incubation. Approximately 20–30 follicles of late stage II, early stage III, or late stage III were randomly allocated and incubated with various chemicals and/or CiVP, and GVBD and ovulation rates were determined after a 24 hr incubation. Synthetic CiVP peptide was prepared as previously described (*Kawada et al., 2008*) and used at a final concentration of 5 µM, given that lower concentrations of other peptides are reportedly ineffective in *Ciona* (*Kamiya et al., 2014*). Linear CiVP with the cysteine residues protected by *N*-ethylmaleimide was synthesized under reducing conditions and confirmed using MALDI-TOF mass spectrometry. The MEK inhibitors U0126 (Promega, Tokyo, Japan), Ro-3306 (Abcam, Tokyo, Japan), and MMP-2/MMP-9 inhibitor II (Calbiochem, San Diego, CA) were used at a final concentration of 10 µM, 1 µM, and 20 µM, respectively. The inhibitory effects of these chemicals on *Ciona* enzymes were evaluated using Western blotting (*Figure 4—figure supplement 1A*), immunofluorescence (*Figure 4A*), and activity assays (*Figure 4—figure supplement 3C* and *Figure 5—figure supplement 3*). Three independent sets of MEK-inhibited follicles were collected for subsequent RNA-seq analyses, and another three independent sets were used for qRT-PCR assays.

## RNA extraction, purification, and RNA-seq analyses

Total RNA was extracted from isolated follicles, purified, and treated with TURBO DNase as previously described (*Matsubara et al., 2017*). A total of 500 ng or 150 ng of quality-verified RNA from sieved follicles or further purified follicles, respectively, was subjected to RNA-seq analysis using a HiSeq1500 instrument (Illumina, San Diego, CA) in the rapid mode, as previously described

**Table 3.** Primer sets used in this study

| Name | Accession no. | Forward | Reverse |
|---|---|---|---|
| qRT-PCR | | | |
| *CiVpr* | KH.C9.885 | ATGCACCGTGTCCGAAATG | CAAAGCGACCAGGACACAAG |
| *CiErk1/2* | KH.L153.20 | TTATGTCTCTGCCGAACAAGC | AAGGTCAGCATACGGTCCAA |
| *CiMmp2/9/13* | KH.L76.4 | GCTTAATTGGCTGCGATCCA | TCGTCTCCATAGTGACATCGG |
| *CiUbac1* | KH.L133.5 | ACACGCCGATGATTCAAGTG | GGTGAGTCGGGTTTGGTTTG |
| Probes for ISH | | | |
| *CiVpr* | KH.C9.885 | CCTGCAGCGACAAATACGAA | GGAACAACTGTGGTGTGGAC |
| *CiErk1/2* | KH.L153.20 | TAAGAGCTCCCACACTTGCA | GCATGATTTCGGGAGCTCTG |

DOI: https://doi.org/10.7554/eLife.49062.029

(*Kawada et al., 2017*; *Kurihara et al., 2017*). The resultant reads were aligned to the *Ciona* cDNA library, which was downloaded from the ghost database (http://ghost.zool.kyoto-u.ac.jp/cgi-bin/gb2/gbrowse/kh/). The expression level of each gene was calculated as gene-specific reads per kilobase per million total reads (RPKM). Total reads, mapping rates, and accession numbers are summarized in *Tables 1* and *2*. Scatter plots are shown in *Figure 2—figure supplement 1* and *Figure 4—figure supplement 2*, and DEG profiles are shown in *Supplementary files 1* and *2*.

## qRT-PCR

A 100 ng aliquot of DNase-treated total RNA isolated from *Ciona* follicles was used for first-strand cDNA synthesis. qRT-PCR was performed using a CFX96 Real-time System and SsoAdvanced Universal SYBR Green Supermix (Bio-Rad laboratories, Hercules, CA), as previously described (*Kawada et al., 2017*). The primers used for qRT-PCR analyses are listed in *Table 3*. Gene expression levels were normalized to the expression of the ubiquitin-associated domain containing one gene (*CiUbac1*, KH.L133.5), which was found to be constitutively expressed throughout follicular development according to the RNA-seq analysis.

In situ hybridization cDNA fragments of *CiVpr* (KH.C9.885) and *CiErk1/2* (KH.L153.20) were amplified using cDNA from *Ciona* ovaries and the primers listed in *Table 3*, and the fragments were then inserted into the pCR4-TOPO vector (Thermo Fisher Scientific, Waltham, MA). After the sequences were verified, the vector was linearized using *Not*I or *Pme*I for the synthesis of digoxigenin (DIG)-labeled cRNA probes, as previously described (*Kawada et al., 2011*; *Matsubara et al., 2011*). Isolated *Ciona* ovaries were fixed in 4% paraformaldehyde (PFA), embedded in Super Cryo Embedding Medium (SCEM), and sectioned at 10 μm. Sections were treated with 1 μg/ml proteinase K (Nacalai Tesque, Kyoto, Japan) for 1–2 min at room temperature and re-fixed in the same fixative for 10 min. The samples were acetylated with 0.25% acetic anhydride in 0.1 M triethanolamine/HCl (Nacalai Tesque) (pH 8.0) for 10 min. After pre-hybridization for 1 hr at room temperature in buffer A (50% formamide, 6 × SSPE, 5 × Denhardt's solution, and 0.5 mg/ml yeast transfer RNA) or buffer B (50% formamide, 10 mM Tris-HCl, 1 mM EDTA, 0.6 M NaCl, 0.25% SDS, 1 × Denhardt's solution, 0.2 mg/ml yeast transfer RNA [pH 8.0]), hybridization was performed at 55–60°C for 16 hr in identical buffer containing 100–200 ng/ml DIG-labeled cRNA probes. The slides were washed three times with 0.2 × SSC at 55–60°C for 20 min each, blocked with 1% blocking reagent (Roche, Basel, Switzerland) in DIG buffer 1 (100 mM maleic acid, 150 mM NaCl [pH 7.5]) for 1 hr, and then treated with anti-DIG antibody (Roche, 1:5000) for 30 min. Signals were developed using NBT/BCIP in DIG buffer 3 (100 mM Tris-HCl, 100 mM NaCl, 50 mM MgCl$_2$ [pH 9.5]).

## In vitro fertilization

After CiVP-directed induction of GVBD in late stage II follicles, ovulated oocytes and un-ovulated follicles were incubated at 20°C for an additional 24 hr in fresh ASW. The outer follicular cell layer of un-ovulated follicles was removed by pipetting. Collected oocytes were mixed with *Ciona* sperm (1.38 to 5.01 × 10$^6$ cells/ml) for 10 min and washed with ASW three times. After incubation at 20°C for 16 hr, embryonic development was evaluated.

## Immunofluorescence

Isolated *Ciona* follicles were de-folliculated in ASW containing 0.1% collagenase and 0.1% actinase E for 1 hr with gentle agitation. After washing, the oocytes were treated with 5 μM CiVP for 0, 5, 10, 20, 40, and 60 min and fixed with 3.7% PFA in ASW at 4°C overnight. For MEK inhibition, oocytes were pretreated with 10 μM U0126 for 1 hr before CiVP stimulation. Oocytes were treated with 0.2% Triton X-100% and 0.5% blocking reagent (PerkinElmer Japan Co., Kanagawa, Japan, FP1020) and then incubated with antibody against pERK1/2 (Cell Signaling Technology [CST], Danvers, MA, 9101S) in Can Get Signal Immunostain Solution A (Toyobo, Osaka, Japan) for 1.5 hr and Alexa 546–conjugated anti-rabbit IgG (Thermo, A11035) for 1.5 hr. Signal was observed using a Leica M205 AF fluorescence stereomicroscope and quantified using Fiji software (*Schindelin et al., 2012*). The specificity of the anti-pERK1/2 antibody against pCiErk1/2 was confirmed by Western blotting and immunofluorescence analyses (see below and *Figure 4—figure supplement 1*).

## Western blotting

Isolated stage IV follicles were homogenized in TBS containing 1 × Complete protease inhibitor cocktail (Roche) and 1 × PhosSTOP phosphatase inhibitor cocktail (Roche). Soluble proteins were collected by two rounds of centrifugation (13,000 rpm, 10 min, 4°C), and protein concentration was determined using a Pierce BCA Protein Assay kit (Thermo). A 30 µg aliquot of soluble protein was electrophoresed, blotted onto a PVDF membrane (GE Healthcare, Buckinghamshire, UK), and blocked with Block Ace (DS Pharma Biomedical Co., Ltd., Osaka, Japan). Antibodies against ERK1/2 (1:500, CST, 9102S) and pERK1/2 (1:1000, CST, 9101S) in Can Get Signal solution were used as primary antibodies. Blocking peptide (1:200, CST, 1150S) was pre-incubated with the anti-pERK1/2 antibody to confirm the antibody's specificity. For CiCdk1, 40 µg of soluble protein extracted in 1.5 × extraction buffer (EB, 30 mM HEPES, 30 mM EGTA, 22.5 mM $MgCl_2$, 1.5 mM ATP, 1.5 mM DTT, 1.5 × Complete protease inhibitor cocktail, and 1.5 × PhosSTOP phosphatase inhibitor cocktail) and p13suc1-precipitate were incubated with anti-PSTAIR primary antibody (1:1000, Abcam, ab10345) in TBS containing 0.05% Tween 20 (TBST). In the case of CiMMP2/9/13, anti–MMP-2/9/13 specific antiserum was generated (see below). Soluble *Ciona* ovary protein was homogenized in PBS. Protein secreted by the ovaries was collected from ASW after 24 hr of incubation and concentrated using Amicon Ultracel-30K centrifugal filters (Merck Millipore, Burlington, MA). A blot of 30 µg of soluble and secreted protein was blocked with Blocking One (Nacalai Tesque) and 4% normal donkey serum (Abcam, ab7475) and then incubated with anti–MMP-2/9/13 antiserum (1:3000 in Can Get Signal solution) as the primary antibody. Pre-immune serum (1:3000) or pre-absorbed antiserum (1:3000) with 70 µg of antigen was used as a negative control. After washing with TBST, the blot was incubated with the corresponding HRP-conjugated anti-mouse IgG (1:2000, GE healthcare, NA9310V) or anti-rabbit IgG (1:2000, GE Healthcare, NA9340V) secondary antibody. Signals were developed using ECL substrate (GE Healthcare).

## Enrichment of CiCdk1 and in vitro activity assay

Isolated stage IV follicles were homogenized in 1.5 × EB. Soluble proteins were collected by two rounds of centrifugation (13,000 rpm, 10 min, 4°C), and the protein concentration was determined using a Bradford Protein Assay (Bio-Rad). A total of 5 µg of soluble protein was incubated overnight with 20 µl of p13suc1-sepharose at 4°C with rotation. The p13suc1-sepharose beads were kindly provided by Prof. Masakane Yamashita of Hokkaido University. After washing 3 times with 1 ml of wash buffer (20 mM HEPES, 5 mM EGTA, 15 mM $MgCl_2$, 1 mM DTT, 1 × Complete protease inhibitor cocktail, 1 × PhosSTOP phosphatase inhibitor cocktail, and 0.2% Tween 20 [pH 7.5]), the enzymatic activity of CiCdk1 was measured using a MESACUP Cdc2/Cdk1 Kinase Assay kit (MBL, Nagoya, Japan) according to the manufacturer's instructions, with the exception that the kinase reaction was performed using sepharose with and without 1 µM Ro-3306 at 30°C for 45 min.

## Preparation of recombinant CiMMP2/9/13 and antibody generation

The cDNA fragment corresponding to the active form of CiMMP2/9/13 (KH.L76.4, amino acid sequence of residues 212–792) was amplified using cDNA from the ovary, Prime Star HS DNA polymerase (Takara, Shiga, Japan), and the primer pair 5'-ACAAGAGGTCCATCGTTCGG-3' and 5'-GTCGTAGTTGTCAGTGGTAG-3'. The DNA fragment was cloned into the pCR-Blunt II-TOPO vector (Thermo). DNA sequencing revealed that the insert contained a non-synonymous SNP resulting in the substitution of W with G at residue 252, which was confirmed by two independent RT-PCR analyses. In addition, 42 amino acids were inserted at residue 569: 38 amino acids were aligned with residues 569–606 of the sequence in the NCBI database (XP_009861837.1), with repetitive extension of 4 amino acids (GTGT), indicating successful cloning of the recombinant protein without any frameshifts. The *Eco*RI/*Hind*III fragment was inserted into the identical sites of the pET21a expression vector (Novagen, Madison, WI). Expression of rCiMMP2/9/13 was induced using the Rossetta 2 (DE3) expression system (Novagen). *Escherichia coli* cells were sonicated in 100 mM Tris-HCl (pH 7.5), and soluble proteins were removed by centrifugation. The precipitate was suspended in buffer composed of 50 mM $NaH_2PO_4$, 300 mM NaCl, and 6 M urea and solubilized by sonication. rCiMMP2/9/13 was purified with a vector-derived His-tag at the C-terminus using Ni-NTA agarose (QIAGEN, Hilden, Germany). Purified rCiMMP2/9/13 was concentrated in PBS by ultrafiltration using an Amicon Ultra-centrifugal filter (Merck, Tokyo, Japan). Purified rCiMMP2/9/13 was then injected into rabbits

for immunization (Hokudo Co., Ltd., Hokkaido, Japan). The specificity of the resultant antiserum against CiMMP2/9/13 was confirmed by Western blotting (*Figure 5—figure supplement 2*).

## Immunohistochemistry

*Ciona* ovaries were fixed in Bouin's solution, embedded in SCEM, and sectioned at 10 µm. The resulting slides were dried at 37°C for 1 hr and washed with PBS. The samples were then boiled in 10 mM citrate buffer (pH 6.0) for 5 min using a microwave oven and cooled for 30 min at room temperature. Endogenous peroxidase activity was quenched by incubating the samples with 0.3% hydrogen peroxide for 10 min. The samples were then sequentially blocked for 30 min each with Blocking One (Nacalai Tesque) and normal horse serum of the R.T.U. VECTASTAIN Universal Elite ABC kit (Vector Laboratories, Burlingame, CA). Anti–MMP-2/9/13 antiserum was used as the primary antibody. Immunoreactivity was assessed using an ABC kit according to the manufacturer's instructions.

## In vitro collagenase activity assay

A total of 0.5 µg of rMMP-2/913 was incubated with 5 µg of DQ collagen type I or type IV in 200 µl of reaction buffer (50 mM Tris-HCl, 150 mM NaCl, 5 mM $CaCl_2$) in the presence or absence of 1 µM MMP-2/MMP-9 inhibitor II at 20°C. After 48 hr of incubation, the fluorescence was measured using a FlexStation II (Molecular Devices, Sunnyvale, CA) with the following settings: Ex, 485 nm; Em, 538 nm; cut-off, 495 nm.

## In situ zymography

Isolated late stage III follicles were incubated with 50 µg/ml of DQ collagen type I (Thermo, D12060) or DQ collagen type IV (Thermo, D12052) in ASW at 20°C for 3 hr. The follicles were then washed with ASW three times, and fluorescence was observed using a Leica M205 FA fluorescence stereomicroscope.

## Statistical analysis

Results are shown as the mean ± standard error of the mean (SEM) of at least three independent experiments. Data were analyzed using the Student's *t* test for one or two pairs of samples and one-way ANOVA followed by Tukey's *post hoc* test for multiple samples using Free-JSTAT statistical software (version 22.0E, Masato Sato, Japan). Differences were considered statistically significant at $p < 0.05$.

# Acknowledgements

We gratefully acknowledge Prof. Shigetada Nakanishi and Prof. Takayuki Takahashi for providing fruitful comments regarding the manuscript. We are also grateful to Prof. Masakane Yamashita for sharing p13suc1-sepharose and to the National Bio-Resource Project for providing ascidians. This work was supported in part by grants from the Japan Society for the Promotion of Science to SM (JP16K18581 and JP17J10624) and HS (JP16K07430).

# Additional information

## Funding

| Funder | Grant reference number | Author |
|---|---|---|
| Japan Society for the Promotion of Science | JP16K07430 | Honoo Satake |
| Japan Society for the Promotion of Science | JP16K18581 | Shin Matsubara |
| Japan Society for the Promotion of Science | JP17J10624 | Shin Matsubara |

The funders had no role in study design, data collection and interpretation, or the decision to submit the work for publication.

### Author contributions
Shin Matsubara, Conceptualization, Supervision, Funding acquisition, Validation, Investigation, Writing—original draft, Project administration, Writing—review and editing; Akira Shiraishi, Tsuyoshi Kawada, Data curation, Validation, Investigation, Writing—original draft; Tomohiro Osugi, Data curation, Investigation, Writing—original draft; Honoo Satake, Conceptualization, Supervision, Investigation, Writing—original draft, Project administration, Writing—review and editing

### Author ORCIDs
Shin Matsubara ⓘ https://orcid.org/0000-0002-7794-534X
Akira Shiraishi ⓘ https://orcid.org/0000-0003-3456-3074
Tomohiro Osugi ⓘ https://orcid.org/0000-0001-6987-9576
Honoo Satake ⓘ https://orcid.org/0000-0003-1165-3624

### Decision letter and Author response
Decision letter https://doi.org/10.7554/eLife.49062.034
Author response https://doi.org/10.7554/eLife.49062.035

## Additional files
### Supplementary files
• Supplementary file 1. DEG profiles based on RNA-seq of fractionated follicles. DEGs (upregulated (>2 fold) or downregulated (<0.5 fold) genes) in the indicated stages of follicles are shown in each tab. Gene ID (column A), reads mapped to the cDNA library (column B-K), RPKM (column L-U), ratio (column V-AC), UniProt ID (column AD), homologous protein (column AE), and E-value (column AF) are shown.
DOI: https://doi.org/10.7554/eLife.49062.030

• Supplementary file 2. DEG profiles based on RNA-seq of MEK-inhibited follicles. DEGs (upregulated (>2 fold) or downregulated (<0.5 fold) genes) in early stage III follicles following MEK-inhibition for 24 hr are shown. Gene ID (column A), reads mapped to the cDNA library (column B-G), RPKM (column H-M), ratio (column N-P), UniProt ID (column Q), homologous protein (column R), and E-value (column S) are shown.
DOI: https://doi.org/10.7554/eLife.49062.031

• Transparent reporting form
DOI: https://doi.org/10.7554/eLife.49062.032

### Data availability
All data generated or analyzed in this study are included in the manuscript and supporting files. Accession numbers of RNA-seq data in this study are described in Table 1 and Table 2. All RNA seq-data are provided in Supplementary files 1 and 4.

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
