## [Decision Letter]

Thank you for submitting your article "The regulation of oocyte maturation and ovulation in proto-vertebrates" for consideration by *eLife*. Your article has been reviewed by three peer reviewers, and the evaluation has been overseen by a Reviewing Editor and Marianne Bronner as the Senior Editor. The following individuals involved in review of your submission have agreed to reveal their identity: Elisabetta Tosti (Reviewer #1); Yutaka Satou (Reviewer #2).

The reviewers have discussed the reviews with one another and the Reviewing Editor has drafted this decision to help you prepare a revised submission.

Summary:

This study provides significant advancement in understanding of how oocyte maturation and ovulation are controlled in an ascidian, *Ciona intestinalis*. Studying oocyte maturation and ovulation in ascidians may provide important insights into how oocyte development evolved. The reviewers are overall very positive and provided comments that can be addressed with straightforward revisions (they are provided as individual comments).

Reviewer #1:

This paper describes the induction of maturation and ovulation by neuropeptides and the related downstream pathways involving MPF mobilization in the ascidian *Ciona robusta*. The manuscript is interesting and provides new evidences on a topic which needs still to be clarified. The experimental design is well conducted and the results are supported by a solid statistic.

I found some major points to be revised:

Abstract: reports too many literature information but poor description of the present data. Modify it accordingly.

Introduction: oocyte maturation is underlined by meiosis. Authors do not well describe the scientific background of their experimental work. An effort should be made to introduce basic concepts as: the nuclear and cytoplasmic maturation; the meiotic arrest at different stages along the species and the recognized GVBD inducers in the most studied model species. (These are poorly described in the fourth paragraph of the Discussion).

Introduction, fourth paragraph: quote Brunetti et al., 2015.

In Materials and methods authors should describe in details the anatomy of reproductive apparatus as they do in Figure 7 to let the readership understand the sequence of maturation in the ovary and ovulation in the oviduct. This is necessary to not confuse the ovulatory process with the following spawning in the sea water.

In the subsection “Western blotting”, quote the temperature and quote the concentration of spermatozoa added.

In conclusion this paper is well written and adds a new contribute the knowledge of oocyte maturation trigger, meiosis resumption and the consequent ovulation in ascidians. Of importance is also the evolutionary implication due to the proved relationship between ascidians and vertebrates. I support the publication on *eLife*

Reviewer #2:

Matsubara et al. reported how oocyte maturation and ovulation are controlled in an ascidian, *Ciona intestinalis*. First they described how oocyte maturation and ovulation proceed in a macroscopic view. Then they revealed that vasopressin, MAPK pathway, and MMP were involved in this control at the molecular level. Their success will probably be based on development of a simple but effective method to fractionate follicles, because it enabled them to do molecular biological assays. Oocyte maturation and ovulation are fundamental biological processes for animals. However, the regulatory mechanism may be very different between vertebrates and other animals. This raises a question how the vertebrate system emerged during evolution. To answer this question, the ascidian system is promising, as they indeed demonstrated. This is because ascidians are the closest invertebrate relative of vertebrates. In addition, the system they revealed in the present study may provide leads for studies in vertebrate systems, because it is possible that a similar mechanism also works in vertebrates. The experimental results shown in this manuscript was clear and sufficient for supporting their conclusions. I think that this manuscript should be published (I have no substantive concerns).

Reviewer #3:

This paper describes experiments that demonstrate that a neuropeptide of the vasopressin/oxytocin family member in ascidians is involved in ovulation and oocyte maturation. They further implicate ERK signaling and MPF activity in these processes, and show that MMP activity may be cleaving collagen downstream of these pathways. They make a number of important technical advances, including observing ovulation in ascidians, fractionating stages of maturing oocytes by size, and localizing collagenase activity in situ.

While the results are solid, and they make considerable progress elucidating the mechanisms of ovulation and maturation in this species, the main conclusions do not make clear, broad implications, and thus read like an incremental advance. They will likely be of limited interest for most readers of *eLife*.

The impact of the work could be enhanced by framing the results in a more general way, and making the evolutionary implications more explicit. The origin of the HPG axis in vertebrates is an important vertebrate innovation. In principle, the results presented here could add to our understanding of its origin, by helping to establish the likely ancestral character state at the base of the vertebrates. From this perspective, it would help if they were clearer about how their findings relate to what has been found in other invertebrates. Do other invertebrates have VP/OT-type peptides? Are the neuropeptides that have been implicated in oocyte maturation in other invertebrates similar in any way to peptide and receptor implicated here? It seems like more can be said about what their work tells us about metazoan oocyte maturation in general.

Similarly, they need to be clearer about how their results relate to what is known in vertebrates. It is very interesting that fish use a VP-related gene for oocyte maturation and ovulation, and this suggests a hypothesis about how their data relates to vertebrates, as they point out. However, this argument could be made more directly. They point out that mouse VP receptor genes are expressed in the ovary, but they do not state whether existing functional studies of VP in mouse would have revealed a role in this organ; it would help to add this if it is true and if they are implying that such a role might exist.

As an aside, is the fish vasotocin peptide similarly related to vasopressin and oxytocin in mammals? Might it represent the ancestral gene these two peptides in mammals? Again, clarification of the distribution of vasopressin and oxytocin peptides across animals might help make the story clearer.

---

## [Author Response]

Reviewer #1:

[…] I found some major points to be revised:Abstract: reports too many literature information but poor description of the present data. Modify it accordingly.

We have revised the Abstract to better describe our data, as indicated below. In addition, please note that the word “proto-vertebrate” and the phrase “conserved prototypic regulatory systems” have been removed in the revised version.

“Ascidians are the closest living relatives of vertebrates, and their study is important for understanding the evolutionary processes of oocyte maturation and ovulation. […] This is the first demonstration of essential pathways regulating oocyte maturation and ovulation in ascidians and will facilitate investigations of the evolutionary process of peptidergic regulation of oocyte maturation and ovulation throughout the phylum Chordata.”

Introduction: oocyte maturation is underlined by meiosis. Authors do not well describe the scientific background of their experimental work. An effort should be made to introduce basic concepts as: the nuclear and cytoplasmic maturation; the meiotic arrest at different stages along the species and the recognized GVBD inducers in the most studied model species. (These are poorly described in the fourth paragraph of the Discussion).

Thank you for your constructive suggestion. In the Introduction section of the revised version, we have described in greater detail the background of oocyte maturation in relation to nuclear and cytoplasmic maturation, the species-specific stage of second meiotic arrest, inducers of oocyte maturation/ovulation in other invertebrates, and the similar and unique properties of the molecular mechanism underlying GVBD induction between *Ciona* and other animals.

“Meiosis in animal oocytes is arrested at prophase of the first division (ProI). Hormonal stimulation triggers the resumption of meiosis in most vertebrates and invertebrates, and oocytes undergo nuclear maturation upon the onset of nuclear disassembly (i.e., germinal vesicle breakdown [GVBD]). […] Mature oocytes are then ovulated via proteolytic degradation of the follicle walls (Richards and Ascoli, 2018; Takahashi et al., 2018; Richards and Pangas, 2010).”

Introduction, fourth paragraph: quote Brunetti et al., 2015.

We have cited Brunetti et al., 2015 in the reference list.

In Materials and methods authors should describe in details the anatomy of reproductive apparatus as they do in Figure 7 to let the readership understand the sequence of maturation in the ovary and ovulation in the oviduct. This is necessary to not confuse the ovulatory process with the following spawning in the sea water.

Thank you for your comments. We have illustrated the anatomic structure of the *Ciona* ovary in revised Figure 7A and B, and we have added appropriate explanations in the Materials and methods section.

“Mature oocytes in the ovary are extruded into the ovarian tube (ovulated) and stored in the oviduct until spawning (Figure 7A and B, Sugino et al., 1990; Kawamura et al., 2011).”

In the subsection “Western blotting”, quote the temperature and quote the concentration of spermatozoa added.

We have added the temperature of follicle/embryo incubation and the number of spermatozoa in IVF, as follows:

“After CiVP-directed induction of GVBD in late stage II follicles, ovulated oocytes and un-ovulated follicles were incubated at 20°C for an additional 24 h in fresh ASW. […] After incubation at 20°C for 16 h, embryonic development was evaluated.”

In conclusion this paper is well written and adds a new contribute the knowledge of oocyte maturation trigger, meiosis resumption and the consequent ovulation in ascidians. Of importance is also the evolutionary implication due to the proved relationship between ascidians and vertebrates. I support the publication on eLife

Reviewer #3:

[…] While the results are solid, and they make considerable progress elucidating the mechanisms of ovulation and maturation in this species, the main conclusions do not make clear, broad implications, and thus read like an incremental advance. They will likely be of limited interest for most readers of eLife.The impact of the work could be enhanced by framing the results in a more general way, and making the evolutionary implications more explicit. The origin of the HPG axis in vertebrates is an important vertebrate innovation. In principle, the results presented here could add to our understanding of its origin, by helping to establish the likely ancestral character state at the base of the vertebrates. From this perspective, it would help if they were clearer about how their findings relate to what has been found in other invertebrates. Do other invertebrates have VP/OT-type peptides? Are the neuropeptides that have been implicated in oocyte maturation in other invertebrates similar in any way to peptide and receptor implicated here? It seems like more can be said about what their work tells us about metazoan oocyte maturation in general.

Thank you for your constructive comments. First, we address the reviewer’s question. VP/OT family peptides are conserved in vertebrates and many invertebrates, including ascidians, amphioxus, echinoderms, annelids, molluscs, and nematodes. Moreover, some insects (e.g., red beetles) possess a VP/OT family peptide, whereas other insects (e.g., fruit flies) lack such peptides (Banerjee et al., 2017; Stoop, 2012; Donaldoson and Young, 2008). These peptides, with some exceptions, completely share Cys^1^, Asn^5^, Cys^6^, Pro^7^, and Gly^9^. In mammals, VP and OT are classified based on the amino acid residue present at position 8; the VP family peptides contain a basic amino acid, and the OT family peptides contain a neutral amino acid at this position. The respective homologs are conserved in non-mammalian vertebrates. Non-mammalian vertebrate VP homologs are designated as vasotocin (VT). Moreover, non-mammalian tetrapod and fish OT homologs are designated mesotocin (MT) and isotocin (IT), respectively. Notably, only VT is present in the lowest vertebrate agnathans. Most invertebrates also possess a single VP/OT family peptide. In addition, some invertebrate VP/OT family peptides harbor a basic amino acid, whereas others harbor a non-basic amino acid. On the other hand, the structural organization is conserved in all precursors of VP/OT family peptides: a signal peptide region, a single VP/OT family peptide sequence, and a neurophysin domain that features multiple cysteines (Banerjee et al., 2017; Stoop, 2012; Donaldson and Young, 2008). Collectively, VP/OT family peptides are thought to have emerged in an ancestral stem group of vertebrates and invertebrates ~600 million years ago (Banerjee et al., 2017; Stoop, 2012; Donaldoson and Young, 2008), leading to the occurrence of the VP and OT family during the evolution of gnathostomates via gene duplication, while classification of invertebrate VP/OT family peptides into the VP or OT family on the basis of the residue at position 8 is impossible (Banerjee et al., 2017; Stoop, 2012; Donaldson and Young, 2008).

Next, we address the question regarding the biological roles of VP/OT family peptides and the regulation of oocyte maturation and ovulation by peptides. To date, multiple biological functions of these peptides have been reported in various organisms. However, the present paper shows for the first time that an OT/VP family peptide regulates oocyte maturation and ovulation in an organism, as described in the original manuscript. Instead, a few species-specific peptides have been shown to be responsible for oocyte maturation and ovulation in other invertebrates such as jellyfish, starfish, and sea cucumber, and thus, we mentioned them in the Introduction and Discussion sections of the revised manuscript, indicating that inducible factors and the molecular mechanisms underlying oocyte maturation and ovulation vary among invertebrate species. Additionally, in jellyfish, direct induction of oocytes by MIH (Takeda et al., 2018) is similar to the direct activity of CiVP on oocytes rather than the two-step induction by starfish relaxin-like peptide via 1-methyladenosine (a direct inducer of oocyte maturation and ovulation in starfish) produced in and secreted from follicular cells (Mita et al., 2009). These points have also been included in the revised manuscript, as shown below:

In the Introduction section:

“The neuropeptide W/RPRPamide in jellyfish directly induces oocyte maturation and spawning (Takeda et al., 2018). […] These findings demonstrate that various neuropeptides are responsible for triggering oocyte maturation and ovulation in invertebrates, and suggest that oocyte maturation and ovulation and their underlying molecular mechanisms are regulated in both a species-specific and evolutionarily conserved fashion.”

In the Discussion section:

“The vertebrate OT and VP families are thought to have emerged from a common ancestral gene via gene duplication during the evolution of gnathostomates, and invertebrates and jawless vertebrates have been shown to possess a single VP/OT family peptide (Banerjee et al., 2017; Stoop, 2012; Donaldoson and Young, 2008). […] VT was also reported to be involved in oocyte maturation and ovulation in catfish (Singh and Joy, 2011; Joy and Chaube, 2015).”

Similarly, they need to be clearer about how their results relate to what is known in vertebrates. It is very interesting that fish use a VP-related gene for oocyte maturation and ovulation, and this suggests a hypothesis about how their data relates to vertebrates, as they point out. However, this argument could be made more directly. They point out that mouse VP receptor genes are expressed in the ovary, but they do not state whether existing functional studies of VP in mouse would have revealed a role in this organ; it would help to add this if it is true and if they are implying that such a role might exist.

As described in the original manuscript, VT, the fish VP homolog, was shown to be involved in oocyte maturation and ovulation. This is the only previous study demonstrating the activity of VP/OT family peptides in induction of oocyte maturation and ovulation. However, as stated in the original version as well, we have verified that the regulation of oocyte maturation and ovulation via CiVP is distinct from that mediated by catfish VT. In addition, animals genetically deficient in VPR and OTR were generated, but no phenotypic defects in oocyte maturation and ovulation have been reported. Considering these previous findings and the present data, we rephrased the corresponding text in the Discussion section, as shown below:

“However, VT-induced oocyte maturation and ovulation are mediated primarily via the VT receptor in follicular cells and following sexual steroidogenesis (Singh and Joy, 2011; Joy and Chaube, 2015; Rewat et al., 2018). […] Consequently, elucidating the biological roles of VP/OT family peptides in the ovary and in the process of VPergic oocyte maturation and ovulation evolution awaits further investigations.”

As an aside, is the fish vasotocin peptide similarly related to vasopressin and oxytocin in mammals? Might it represent the ancestral gene these two peptides in mammals? Again, clarification of the distribution of vasopressin and oxytocin peptides across animals might help make the story clearer.

As described in the original manuscript, VT, the fish VP homolog, was shown to be involved in oocyte maturation and ovulation. This is the only previous study demonstrating the activity of VP/OT family peptides in induction of oocyte maturation and ovulation. However, as stated in the original version as well, we have verified that the regulation of oocyte maturation and ovulation via CiVP is distinct from that mediated by catfish VT. In addition, animals genetically deficient in VPR and OTR were generated, but no phenotypic defects in oocyte maturation and ovulation have been reported. Considering these previous findings and the present data, we rephrased the corresponding text in the Discussion section, as shown below:

“However, VT-induced oocyte maturation and ovulation are mediated primarily via the VT receptor in follicular cells and following sexual steroidogenesis (Singh and Joy, 2011; Joy and Chaube, 2015; Rewat et al., 2018). […] Consequently, elucidating the biological roles of VP/OT family peptides in the ovary and in the process of VPergic oocyte maturation and ovulation evolution awaits further investigations.”